EMBO
reports

# Extensive flavivirus E trimer breathing accompanies stem zippering of the post-fusion hairpin

Iris Medits[1,‡], Marie-Christine Vaney[2,‡], Alexander Rouvinski[2,†], Martial Rey[3], Julia Chamot-Rooke[3], Felix A Rey[2,*] (iD), Franz X Heinz[1,**] (iD) & Karin Stiasny[1,***] (iD)

## Abstract

**Flaviviruses enter cells by fusion with endosomal membranes through a rearrangement of the envelope protein E, a class II membrane fusion protein, into fusogenic trimers. The rod-like E subunits bend into "hairpins" to bring the fusion loops next to the C-terminal transmembrane (TM) anchors, with the TM-proximal "stem" element zippering the E trimer to force apposition of the membranes. The structure of the complete class II trimeric hairpin is known for phleboviruses but not for flaviviruses, for which the stem is only partially resolved. Here, we performed comparative analyses of E-protein trimers from the tick-borne encephalitis flavivirus with sequential stem truncations. Our thermostability and antibody-binding data suggest that the stem "zipper" ends at a characteristic flavivirus conserved sequence (CS) that cloaks the fusion loops, with the downstream segment not contributing to trimer stability. We further identified a highly dynamic behavior of E trimers C-terminally truncated upstream the CS, which, unlike fully stem-zippered trimers, undergo rapid deuterium exchange at the trimer interface. These results thus identify important "breathing" intermediates in the E-protein-driven membrane fusion process.**

**Keywords** class II fusion protein; dynamic behavior; flavivirus E trimer; fusion intermediate; membrane fusion; stem zippering

**Subject Categories** Microbiology, Virology & Host Pathogen Interaction; Structural Biology

## Introduction

Flaviviruses cause some of the most devastating arthropod-borne diseases around the world, including dengue, yellow fever, Zika, Japanese encephalitis, West Nile encephalitis, and tick-borne encephalitis (TBE) [1]. Their emergence and spread into new geographical areas pose substantial challenges for international public health. Although important progress has been made to understand the structural organization of the flavivirus particles and the functions of their envelope proteins during the viral life cycle, many essential aspects still remain unexplored. In particular, the detailed molecular interactions of the flavivirus envelope (E) protein to drive membrane fusion for entry into cells have remained elusive. Understanding these details could provide new targets for the development of specific treatments against flaviviruses in general.

Flavivirus particles are lipid-enveloped, pH-sensitive macromolecular machines containing a capsid protein C and two transmembrane glycoproteins, prM and E. The virion assembles on the membrane of the endoplasmic reticulum (ER) and buds into the ER lumen as an immature particle containing prM/E heterodimers [2]. Key to the infectivity of the virion is its reactivity to low pH, which leads to a major particle reorganization twice during its life cycle. The first low-pH exposure occurs in the trans-Golgi network (TGN) of the cell into which immature particles are transported during exocytosis, allowing the proteolytic cleavage of prM into a peripheral "pr" domain and the membrane-anchored "M" subunit to generate activated, mature infectious virions containing M/E complexes [3]. The second low-pH exposure occurs in the endosome, when the virus enters a new cell by endocytosis. Here, the E protein mediates fusion of the viral envelope with the endosomal membrane, thereby releasing the genome into the cytoplasm [4].

The E protein is about 500 amino acids (aa) long, with an ectodomain formed by the ~ 400 N-terminal aa and folded as three $\beta$-sheet rich domains I, II, and III (Fig 1). The ectodomain is followed by a flexible region termed "stem" that tethers it to the C-terminal transmembrane (TM) helices. The stem plays a crucial role in controlling the pH-dependent transitions of the particle and, in particular, in the membrane fusion step driven by protein E. As revealed by the near-atomic resolution cryo-EM structures of mature virions of several flaviviruses (including Zika, dengue, Japanese encephalitis, and

---

1   Center for Virology, Medical University of Vienna, Vienna, Austria
2   Unité de Virologie Structurale, Institut Pasteur, CNRS UMR 3569 Virologie, Paris, France
3   Unité de Spectrométrie de Masse pour la Biologie, Institut Pasteur, CNRS USR 2000, Paris, France
    *Corresponding author. Tel: +33 1 4568 8563, E-mail: felix.rey@pasteur.fr
    **Corresponding author. Tel: +43 1 40160 65500, E-mail: franz.x.heinz@meduniwien.ac.at
    ***Corresponding author. Tel: +43 1 40160 65505, E-mail: karin.stiasny@meduniwien.ac.at
    ‡These authors contributed equally to this work
    †Present address: Department of Microbiology and Molecular Genetics, Institute for Medical Research Israel-Canada, The Kuvin Center for the Study of Infectious and Tropical Diseases, The Hebrew University of Jerusalem, Jerusalem, Israel

---

TBE viruses) [5–11], the stem forms three mostly amphipathic α-helices (H1, H2, and H3) that are oriented tangentially to the particle and are sandwiched between the viral membrane and the E-protein ectodomain layer (Fig 1A). Although the aa sequence of the stem is relatively variable across flaviviruses, a short segment upstream of H3 features a highly conserved sequence (CS) (Fig 1).

The E protein is maintained in a metastable, dimeric pre-fusion state at the surface of mature particles at neutral pH. Specific pH-dependent interactions of its ectodomain with both the H1 helix of the stem and with M (Fig 1A) have been postulated as important to hold it in this metastable state [8,9]. Exposure to the low-pH environment of the endosome weakens these inter- and intra-molecular interactions, contributing to E dimer dissociation to initiate fusion. The stem has also been proposed to play a further role in facilitating an outward extension of E monomers during entry [12], thereby projecting the fusion loop (FL) to insert into target membranes and allow E homotrimerization, both required to initiate fusion. E trimerization is thought to occur upon FL insertion into a target membrane, and to take place while the E subunits are in an extended conformation, bridging the two membranes at a distance of about 150 Å. The individual E subunits of the transient extended trimers then bend in the middle, collapsing into an overall "hairpin" conformation that brings domain III and the stem to the sides of a core trimer formed by the domain I/II moieties. The overall conformational change thus forces the juxtaposition of target and viral membranes for inter-bilayer lipid contacts, thereby catalyzing their fusion into a single membrane [4,13].

Although structural details of C-terminally truncated forms of the post-fusion E trimer have been reported for several flaviviruses [14–18], the interactions of the stem with the trimer core have remained elusive [19]. Current fusion models postulate that the stem zippers together neighboring domains II in the trimer, providing part of the driving force for membrane merger [4,20]. This model is supported by the observation that stem-derived peptides specifically bind to recombinant E trimers truncated of the stem [21]. These same peptides also inhibit membrane fusion and/or infection by several flaviviruses [21–26]. The precise mechanism of inhibition by these peptides, however, remains controversial [25]. Furthermore, it is unknown what parts of the stem contribute most to the stability of the post-fusion trimer, and key structural details of these interactions are lacking. The available X-ray structure of the dengue virus serotype 1 (DENV1) E trimer [19] is currently the most complete snapshot of a flavivirus E trimer in its post-fusion form. This structure, which was obtained with a construct of E truncated at the CS, did not resolve the C-terminal 18 amino acids leading to the CS. Together with binding data obtained with stem-derived peptides [21,23], the DENV1 E trimer structure led to the proposal that the first nine stem residues (aa 395–403 in DENV1) interact tightly with domain II, that the polypeptide chain would then loop out from the body of the trimer (to explain the disorder of the C-terminal segment in the crystals), and that the distal part of the stem (i.e., residues downstream the CS) would interact with the region of the domain II tip to complete the post-fusion hairpin [19].

Here, we provide evidence for a flavivirus E post-fusion trimer in which stem residues do not loop out but are involved in zippering up to the CS and contribute to the stability of the post-fusion E trimer, suggesting that the amphipathic helix (H3) downstream the CS remains membrane associated during fusion. We further show that incompletely stem-zippered E trimers (i.e., truncated upstream the CS)—and in contrast to fully zippered trimers—sample a broad conformational landscape (or "breath") to expose residues of the trimer interface. These observations indicate that during the membrane fusion reaction, the intermediate E trimers are highly dynamic until formation of the final, fully zippered post-fusion hairpin conformation. Based on these observations, we propose a model in which the H3 helix plays an important role in the final steps of membrane fusion, similar to analogous membrane-proximal helices in the class I and class III fusion proteins. The model is supported by thermostability analyses, differential hydrogen–deuterium exchange rates, and monoclonal antibody (mab) binding data using E trimers of TBE virus (TBEV) truncated at different positions along the stem.

## Results

### X-ray structure of a TBEV sE-448 construct with an internal linker replacing aa 405–426

We define the E stem as the segment running from aa 396 to 448 in TBEV numbering (Fig 1B and Appendix Fig S1), from the end of domain III to the TM segment. A previous structure of the TBEV E trimer ectodomain resulting from trypsin digestion of the virion-solubilized intact E protein had resolved the stem up to residue 401 (sE401v structure, PDB 1URZ, [15]). Following the "looping-out" model proposed for DENV1 by Klein *et al* [19], which suggested a role for sequence elements in the distal stem in zippering, we designed a recombinant soluble TBEV E protein (sE-linker) in which an 8-residue linker (GGGSGGGS) connected Gln404 to Gly427 and the following H3, bypassing most of the proximal stem, i.e., the second half of H1, H2, and the CS (Fig 1C and D and Appendix Fig S1). If the H3 segment is responsible for zippering up the trimer, then an 8-residue linker should be long enough to allow it to make the required zippering interactions with domain II. We also introduced the same mutation in the FL (W101H) described in the DENV1 work [19] into the sE-linker construct (sE-linker* mutant), to enhance solubility and facilitate trimerization at low pH under crystallization conditions in the absence of detergent. We obtained crystals of the rhombohedral space group *H3* diffracting to 2.6 Å resolution (see Materials and Methods). We determined the structure by molecular replacement and refined it to a free R value of 21.7% with correct geometry (Appendix Table S1). The crystals displayed clear electron density except for the region of the tip of domain II (Fig 2A), including the *bcd* β-sheet (bearing the FL in the *cd* loop) and the *ij* β-hairpin, which were not resolved. Nevertheless, the density was clear for the stem up to the last residue before the linker (Gln404, Fig 2B and Appendix Fig S2A), which corresponds to Glu403 in DENV1 E (Fig 2C)—the last residue resolved in that structure [19]. As in the DENV1 sE trimer (Fig 2C), in the TBEV sE-linker* structure (Fig 2B) the stem adopts an extended conformation running within a groove, termed the "αA/B" groove, at the inner surface of domain II of the same chain, formed in between α-helices A and B and bounded by the 3/10 helical turn η3 (see Appendix Fig S1 for the secondary structure nomenclature). The stem side chains alternate between inter- and intra-chain contacts: those of Ser397, Ile399, Val402, and Gln404 interact with the

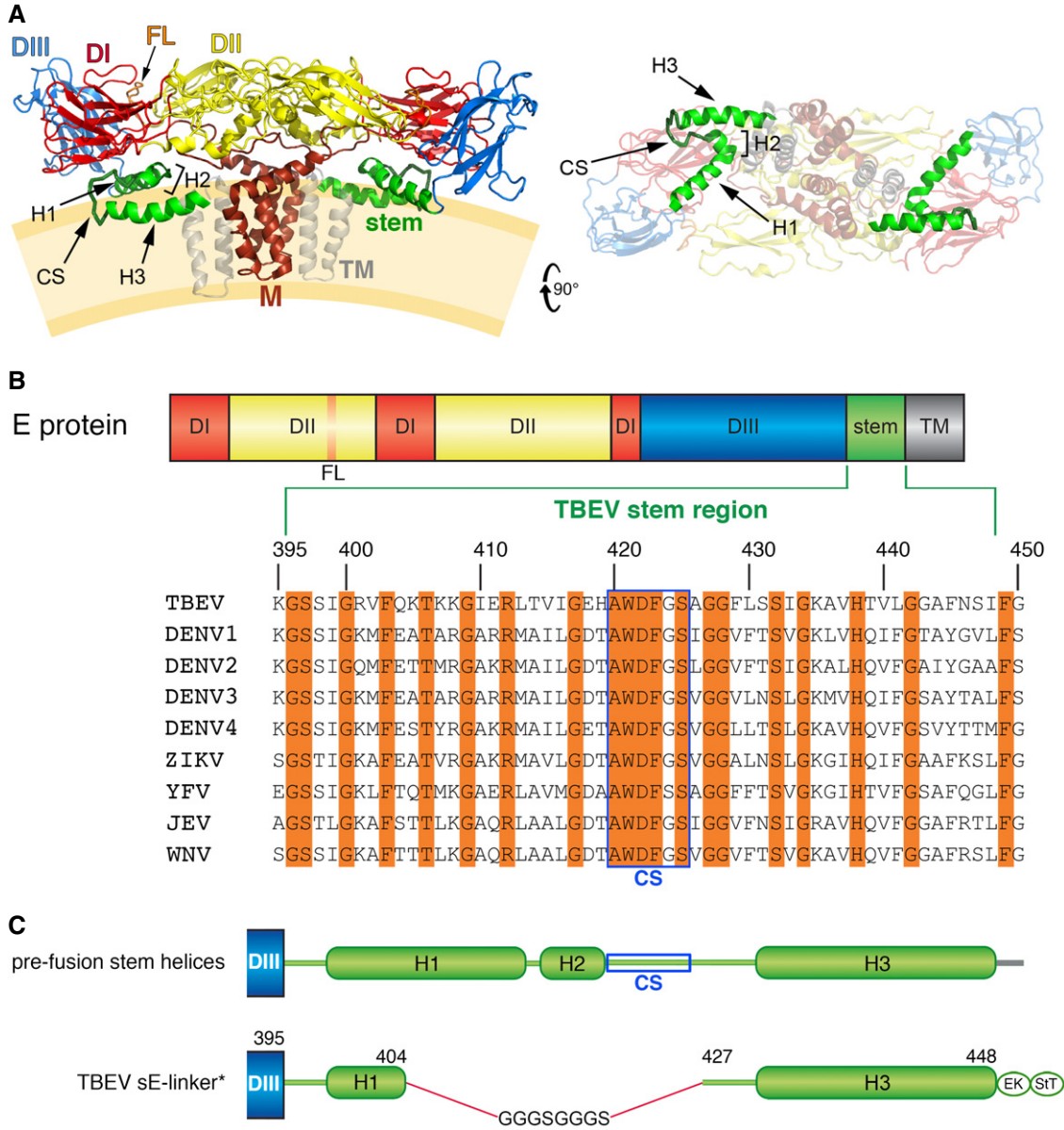

**Figure 1. Structure of the stem in the pre-fusion TBEV (M/E)$_2$ dimer as seen on virions, and construct design.**

A   Ribbon diagrams of the (M/E)$_2$ dimer extracted from the structure on virions (PDB 5O6A) [10], in side (left panel) and bottom (right panel) views with M in brown and E colored according to domains: domain I, red; domain II, yellow; domain III, blue; stem, green; transmembrane (TM) segments, gray; and fusion loop (FL), orange. The α-helices H1, H2, and H3 and the conserved sequence (CS) of the stem are indicated. In the left panel, the viral membrane is diagrammed in light orange.

B   Domain organization of E (color-coded as in A), expanding the stem region (TBEV numbering: aa 395–448) with the corresponding aa sequence alignment (Clustal Omega, https://www.ebi.ac.uk/Tools/msa/clustalo/) using representative flaviviruses (see Material and Methods for the corresponding accession numbers). Strictly conserved aa are highlighted in orange. The CS (located between helices H2 and H3) is framed in blue.

C   Schematic organization of the TBEV stem following domain III (in blue).

D   Schematic organization of the TBEV sE-linker* construct used for crystallization. The deleted segment (405–426) was replaced by an 8-residue linker, and a W101H mutation was introduced in the FL. An enterokinase-specific cleavage site (EK) and a strep-tag (StT) used for affinity purification were added at the C-terminal end of H3.

adjacent subunit, whereas Ser398, Arg401, and Phe403 interact with domain II of the same protomer (the register change from odd to even residues is caused by a bulge of the main chain at Gly400). The domain II region involved in inter-chain contacts with the stem includes the aa 230–238 region between β-strand *h* and helix η3 preceding the *ij* β-hairpin, with the Gln404 side chain hydrogen

bonding the main chain of the neighboring subunit at residue 233 (Fig 2B). Upon structural alignment of the stems of the DENV1 and TBEV structures, the interacting regions of domain II of both structures are also brought into superposition, highlighting the similarity between the interactions. The only differences are due to a three-residue insertion in DENV1 E in the *βh-η3* loop (Appendix Fig S1),

which makes additional contacts and creates a narrower path for the stem (Fig 2C). The stem side chains engaged in multiple interactions in the TBEV structure are Arg401 (the last residue in the TBEV sE401v structure) and Phe403, which is buried in a hydrophobic pocket of domain II (Figs 2B and EV1B), making the same pattern of interactions observed for the corresponding Lys400 and Phe402 side chains in the DENV1 structure (Fig 2C).

The observed disorder at the tip of domain II in the crystals (Fig 2A) was intriguing, as this region was clearly ordered in the TBEV sE401v crystals [15]. The C-terminal extension in the sE-linker construct (Fig 1D) could potentially induce structural instability in this region, affecting the tip of domain II speculated to be in contact with the H3 region [19]. Examination of the crystal packing showed that the previously determined sE401v structure (PDB 1URZ) superposed strikingly well to the ordered part of the sE-linker* structure, but that the tips of domain II, if they remained unchanged, would clash with crystallographically related trimers in the crystal (Appendix Fig S2). We sampled the diffraction of multiple crystals of the TBEV sE-linker* protein and found that they all belonged to the same crystal form but were not isomorphous to

each other, and had a solvent content ranging between 42% and 47% of the unit cell volume. The extent of the disorder at the domain II tip appeared to inversely correlate with the solvent content of the crystal. The three-dimensional lattice of the sE-linker* crystals is much more compact than that observed in the TBEV sE401v crystals, which had a solvent content of 57% [15], even though the crystallization conditions were similar (Appendix Table S1). The crystal packing contacts made by the bottom half of the trimer (Appendix Fig S2) could thus be energetically favorable and drive crystallization at the expense of inducing disorder at the trimer tip, potentially explaining the observed disorder in this region.

### The αA/B groove accommodates the N-terminal M segment on virions

The 3.9 Å-resolution cryo-EM reconstruction of mature TBEV particles [10] showed that the N-terminal segment of M runs in an extended conformation underneath the E ectodomain (Figs 3A and EV1C), along the same hydrophobic groove that accommodates the

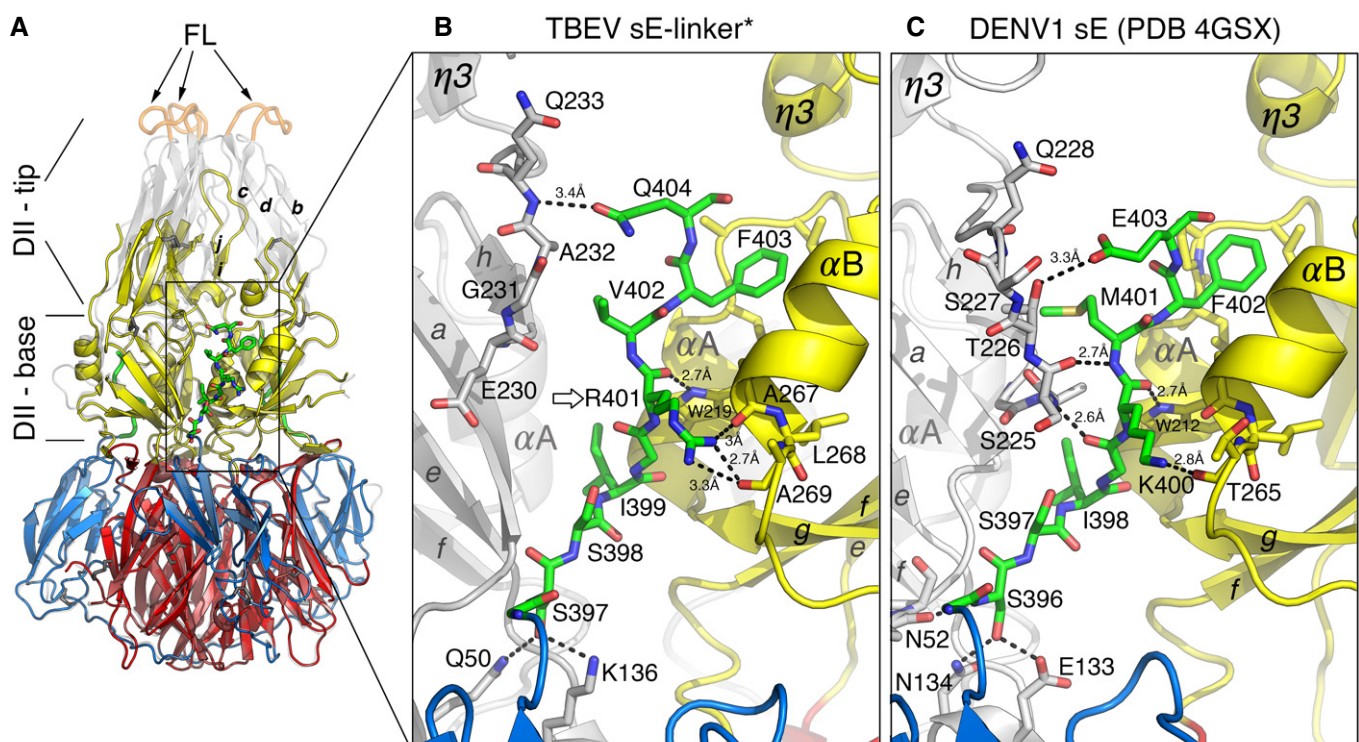

**Figure 2.  X-ray structure of the TBEV sE-linker* trimer.**

A   The structure of the sE-linker* trimer is displayed as ribbons colored by domains as in Fig 1. The previously determined structure of the TBEV sE401v trimer (PDB 1URZ, [15]) is superposed in light gray with fusion loops (FL) in orange, highlighting the disordered domain II tip in the new structure. The two subdomains of domain II, base and tip, are indicated. Disulfide bonds are shown as dark gray sticks. The residues of the stem are displayed as sticks color-coded according to atom type, with carbon atoms green.

B   Close-up view of the sE-linker* trimer with one subunit colored as in (A) and the others in gray. The segment connecting helix αA to the η3 3/10 helical turn (see Appendix Fig S1) of the adjacent subunit is shown as sticks colored according to atom type with carbon atoms light gray. Broken lines mark polar interactions of the stem with surrounding residues (with distances ranging between 2.7 and 3.4 Å). An open arrow marks the C-terminal end of the sE401v trimer structure reported previously (PDB 1URZ).

C   For comparison, the same region of the stem is represented for the DENV-1 sE structure (PDB 4GSX, [19]), using the same color scheme.

stem in the post-fusion trimer but in the opposite direction (Fig 3A, middle panel and Fig EV1B and C). The side chain of residue 4, which conserves its aliphatic character (Val, Leu or Ile) in the M protein of all flaviviruses (Appendix Fig S1), inserts into the same pocket of domain II as Phe403 of the stem (Fig 3A, top-right panel). The indole group of the strictly conserved E Trp219 makes a hydrogen bond with the main-chain carbonyl at position 401 of the stem in the post-fusion form, and with the main-chain carbonyl of M residue 5 in the pre-fusion form (Fig 3A, bottom-right panel). These M–E interactions are conserved and have been visualized in all flaviviruses for which the structure of the mature virion is known (Fig EV2). Importantly, in all cases, the highly conserved M His7 and E His216 (in TBEV numbering) face each other at the N-terminal end of αA as described previously [8,10] (Figs 3A and EV2, right panels), indicating that the M–E interaction is prone to become unstable at acidic pH, upon protonation of the imidazole rings. Indeed, when we detergent-solubilized the E protein from virions at neutral/alkaline pH, an important fraction of M remained associated with the E dimer (Fig 3B), but upon treatment at pH 6 the two proteins separated and E sedimented as a post-fusion trimer detached from M (Fig 3B). The observed low-pH-induced dissociation of the M–E interactions in the pre-fusion form highlights a structural role of M, stabilizing the pre-fusion form of E at neutral pH. The release of the interactions of the E ectodomain with M, as well as with the E stem, thus appears essential for the E swiveling motion that projects the FL against the endosomal membrane to initiate fusion.

### Generation of TBEV E proteins with different stem lengths

Because the structure of the sE-linker* trimer did not provide information on the putative interactions of the stem downstream residue 404, we sought indirect information on the effect of this segment by assessing the differential thermostability, deuterium exchange rate, and reactivity with various mabs of TBEV E trimers ending at different positions along the stem.

We designed the constructs for recombinant protein expression in *Drosophila* cells based on the stem elements defined in the high-resolution cryo-EM structure of the TBEV particle [10] as follows (Fig 4A): sE400r (the "r" at the end of the construct name indicates that it is a recombinant protein and carries a C-terminal strep-tag), sE401r, sE404r (containing part of H1), sE412r (containing H1), sE419r (containing H1 and H2), sE428r (containing H1, H2, and the CS), sE448r (containing the whole stem), and the linker construct sE-linker (albeit without the W101H mutation present in the corresponding protein used for crystallization and structure determination—Fig 1D). In addition, we included in the analyses the full-length E trimer containing the whole stem-anchor region (termed Ev, isolated from low-pH-treated purified virions; Fig 4A), and the virion-derived stem-truncated sE401v trimer (Fig 4A) previously used for structure determination by X-ray crystallography [15,27].

We determined the oligomeric state of the various E proteins by sedimentation analysis and chemical cross-linking. sE448r was secreted directly as a trimer, as described previously [28], but all other recombinant proteins and the virion-derived sE401v had to be converted into trimers by exposure to acidic pH in the presence of liposomes, as described previously ([15,29], Materials and Methods). Sedimentation analysis and chemical cross-linking

experiments revealed high yields of trimers in all instances (Appendix Fig S3), with the exception of sE400r (Appendix Fig S3A). We thus conclude that the single amino acid Arg401 plays a crucial role in the formation of stable trimers, in line with the multiple interactions made by its side chain in the X-ray structures (Fig 2B and [15]). Since we could not obtain sE400r trimers, we did not pursue further studies with this construct.

### Influence of stem length on trimer stability

Because the amount of E trimers detergent-solubilized from liposomes produced by our procedure was limited, it was not feasible to carry out differential scanning calorimetry experiments to compare the thermostability of the various constructs. We therefore used an alternative approach, which consisted in determining the fraction that resisted a 10-min incubation at 70°C, the highest temperature at which the full-length Ev trimer remained unaffected as monitored by sedimentation analysis (Materials and Methods) [28,30]). We observed a strong reduction of the trimer peak for sE401, in both the recombinant (sE401r) and virion-derived (sE401v) forms, indicating lower trimer stability (Fig 4B). Extending the stem by only three amino acids (sE404r trimers) led to an important increase in trimer thermostability, which remained unaltered by the addition of eight more amino acids (sE412r trimers) (Fig 4B). We found a further small but significant increase for the sE419r trimers, which reached the thermal stability of Ev. Further extensions (to include CS and/or H3 residues) had no additional contribution to trimer stability (sE428r and sE448r trimers, Fig 4B). The sE-linker trimer exhibited an intermediate behavior, being less resistant than the sE404r but more than the sE401r/sE401v trimers.

### Hydrogen–deuterium exchange mass spectrometry

To further assess the effect of stem interactions on the conformation of the E trimer, we carried out comparative hydrogen–deuterium exchange mass spectrometry (HDX-MS) experiments. We examined in parallel the exchange rate of amide hydrogens of the following E-protein constructs: sE404r, sE419r, and sE428r. We observed differences in deuterium uptake between constructs sE404r and sE419r ($D_{404}$–$D_{419}$) and between sE404r and sE428r ($D_{404}$–$D_{428}$) that were significant with a 99% confidence (1-$P$ value > 0.99) in a Student's $t$-test (Figs 5A and EV3). Because we were interested in understanding global deuterium exchange differences, we applied a threshold to eliminate minor variations that could be due to protein preparation and only focused on the main changes. The HDX-MS data covered about 70% of the protein, missing two regions from amino acids 50 to 100, and from 300 to 350, which are rich in cysteine residues and contain several disulfide bonds. While there were no obvious overall deuteration differences between sE419r and sE428r, both of these constructs displayed the same overall HDX difference pattern with respect to sE404r (Fig EV3). The data showed that in sE404r there is a strikingly higher deuterium exchange rate in four regions, spanning amino acids 10–22, 162–175, 188–200, and 344–356, termed regions HDX-*a*, HDX-*b*, HDX-*c*, and HDX-*d* (Fig 5A, right panel). The first two regions map to domain I, the third to the hinge between domains I and II, and the fourth to domain III (Fig 5B–D and Appendix Fig S1). The HDX-*a* region is centered on the $A_0B_0$ loop of domain I, which makes multiple inter-chain

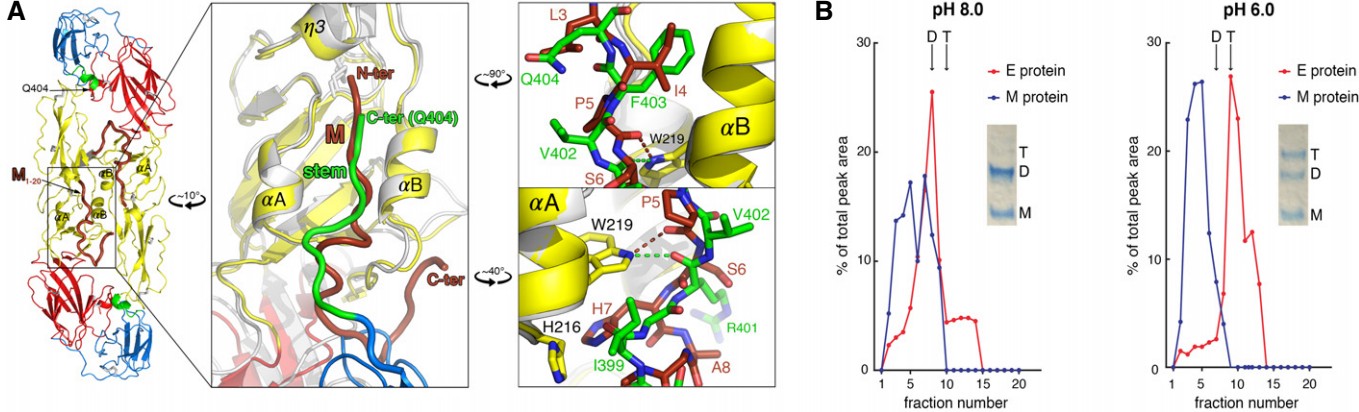

**Figure 3. Ribbon representation of the M–E interaction site on the mature virion, with M inserting into the same groove as the stem in the post-fusion form of E.**

A  Left panel, (M/E)₂ dimer (PDB 5O6A, [10]) colored as in Fig 1. For clarity, residues downstream E Gln404 were deleted, and only residues 1–20 of M are shown in tube representation. The framed region is enlarged in the middle panel, with the pre-fusion E/M heterodimer (E: gray; M: brown) and the post-fusion E trimer (color-coded as domains) superimposed on helices αA and αB, illustrating the common site occupied by M and by the E stem at different stages of the flavivirus cycle. E and M run in opposite directions within the αA/B groove, interacting with the same domain II residues. Right panel: close-ups of the interactions with residues displayed as sticks. Top, residue Ile4 of M takes the place of Phe403 of the stem. The bottom panel illustrates the alternative interactions made by the strictly conserved Trp219 side chain with the main chain of M in pre-fusion and of the stem in post-fusion forms. In both, top and bottom panels, stem and M residues are labeled in green and brown, respectively.

B  pH-dependent dissociation of E and M. TBEV preparations treated at pH 8 (left) and pH 6 (right) were solubilized with Triton X-100 and subjected to sucrose gradient centrifugation. The fractions were analyzed by SDS–PAGE and densitometry. Representative examples of at least two independent experiments are shown. Positions of E dimers (D) and E trimers (T) in the gradients are indicated.

hydrogen bonds about the 3-fold axis at the very bottom of the trimer. Similarly, the HDX-*c* region is centered on a 3/10 helical turn (the η₂ helix) at the hinge between domains I and II, in between β-strands H₀ (domain I) and *f* (domain II). The η₂ helical turn interacts with itself about the 3-fold axis at the trimer center. The HDX-*b* and HDX-*d* regions map away from the 3-fold axis, but are in front of each other in the quaternary structure of the protein, where domain III packs against domain I of an adjacent protomer in the trimer. We interpret the observed decrease in deuterium uptake of these regions in the two longer constructs as reflecting an overall stabilization of the trimer introduced by the presence of the segment spanning amino acids 405–419, in agreement with the increased thermal stability of both sE419r and sE428r constructs. The absence of significant differences in deuterium uptake between the two latter constructs (i.e., differing only by the CS; Fig EV3) is also in agreement with the thermal stability results (Fig 4B).

### Reactivity of truncated and full-length trimers with E-protein-specific mabs

To further explore the effect of the presence of the stem on the E trimer, we carried out comparative antibody binding experiments with various sE constructs (as well as full-length Ev), using mabs for which the epitopes had been previously mapped ([31–34] and also unpublished data) (Fig 6A). We found that the binding of mab A1, which recognizes the FL, is not impaired in constructs with a stem up to residue 419, but is strongly reduced in the presence of a longer stem (Fig 6B), suggesting that residues in the CS and downstream interfere with exposure of the FL epitope at the very tip of the trimer. Mab A2, which had been mapped to bind close to the FL but not at the tip of the trimer, was affected by the presence of more

upstream stem elements, as it exhibited reduced binding with the sE419r construct (Fig 6B), suggesting that stem residues upstream the CS may partially occlude its epitope. We found a similar effect with mab A3 (Fig 6B), which has its epitope in domain II further away from the tip (Fig 6A). It is important to note that the interference with mab binding could be due to direct masking, but could as well be due to an indirect effect, for instance by the stem altering dynamic breathing of domain II in the trimer. As the actual footprint of these mabs on domain II is not known, their epitopes could be partially occluded in the trimer. Such an effect is apparent in the experiments with mab B4, the binding of which was affected by residues in the proximal and distal stem although its epitope maps to domain III, away from the stem-domain II contact site (Fig 6A). A likely explanation for these observations is that a more dynamic behavior of the unzipped shorter trimers allowed for better exposure of the B4 epitope than in the stem-stabilized trimers. Indeed, the B4 epitope maps next to the HDX-*d* region, identified by HDX-MS as having a much higher exchange rate in the sE404r than in the longer constructs (Fig 5). Because the HDX experiments revealed a clear difference in exchange in close proximity to the B4 epitope but not in that of the A1, A2, and A3 mabs, we favor a direct masking effect for the latter, which is also in agreement with the expected stem-binding region in the trimer.

## Discussion

We present here X-ray crystallography data on the TBEV E post-fusion trimer with the stem traced up to residue 404, displaying interactions very similar to those described for its counterpart in the DENV1 E trimer (Fig 2). Although the new structure did not resolve

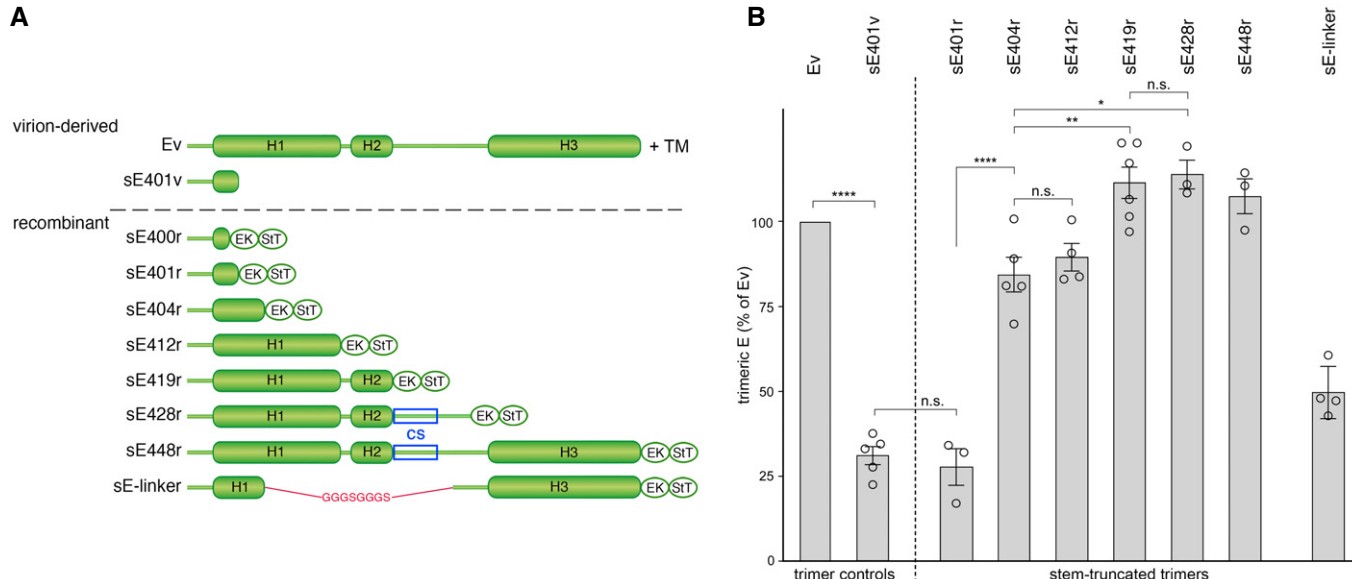

**Figure 4. TBEV E constructs with sequential truncations of the stem and thermostability of the corresponding trimers.**

A All diagrams are aligned at the left and at the same scale, using the full-length E protein (Ev, top line) and sE401v (second line), derived from virions, as reference. Recombinant constructs are shown below the dashed line: sE400r, sE401r, sE404r (containing 6 N-terminal amino acids of H1), sE412r (containing H1), sE419r (containing H1 and H2), sE428r (containing H1, H2, and CS), and sE448r (containing the whole stem). Abbreviations EK and StT indicate heterologous sequences corresponding to an enterokinase-specific cleavage site and a strep-tag used for affinity purification, respectively, added at the C-terminal end of each construct.

B Trimer stability expressed as percentage of the fraction detected in the trimer peak and normalized with respect to the corresponding Ev fraction in sedimentation analyses after incubation at 70°C relative to 37°C (100%). Data are from at least three independent experiments.

Data information: Data shown represent the means ± standard error of the mean ($n$ = 3–6 biological replicates). Statistical significance was determined using one-way ANOVA with Tukey's multiple-comparison test (n.s, not significant; ****$P$ < 0.0001; **$P$ < 0.01; *$P$ < 0.05). Only comparisons described in the text are shown, and complete statistical analyses are summarized in Appendix Table S2.

Source data are available online for this figure.

the downstream stem region, we were nevertheless able to extract significant new information about its contribution to the overall stability of the E trimer. We also provide evidence that suggests the formation of a highly dynamic intermediate of the stem-zippering reaction during viral membrane fusion.

We identified a critical role of residue 401 in trimer formation, as constructs ending at position 400 did not trimerize efficiently (Appendix Fig S3). We also observed that residues 402–404 further provided a strong contribution to trimer thermostability (Fig 4B) and showed that the same domain II groove accommodating this

portion of the stem in the post-fusion form is occupied by M in the pre-fusion form of E on mature virions (Fig 3). Further extension of the sE construct to end at residue 419 resulted in an additional small but significant increase in trimer stability (Fig 4B). Most importantly, this longer construct displayed a strikingly lower deuterium exchange rate as well as a different mab-binding pattern compared to that ending at residue 404 (Figs 5 and 6). The higher deuterium exchange rate of the latter occurred at surface residues involved in inter-subunit interactions within the trimer, which are away from the postulated contact site of the stem with domain II and should be

**Figure 5. Hydrogen–deuterium exchange mass spectrometry (HDX-MS).**

A Difference in deuterium uptake by sE404r and sE419r trimers ($D_{404}$–$D_{419}$). Left panel, deuteration difference for each peptide reported against the 1-$P$ value obtained via Student's $t$-test between the two different constructs. In the right panel, peptides are reported as short horizontal lines, with length and position corresponding to the peptide length and location along the protein sequence ($x$-axis) and with their deuteration delta on the $y$-axis. In both panels, red (positive) and blue (negative) indicate changes above the Student's $t$-test threshold, and in gray those not reaching this threshold. Four peptides, termed HDX-$a$, HDX-$b$, HDX-$c$, and HDX-$d$, were identified as showing a strong increase in deuteration in sE404r compared to sE419r. The same pattern was observed in the comparison between sE404r and sE428r, whereas no difference deemed "significant" by the same test was seen when comparing sE419r with sE428r (see Fig EV3).

B Middle panel: ribbons representation of the sE trimer (PDB 1URZ) colored according to domains with high HDX regions $a$ and $c$ boxed and highlighted in light gray. The central box marks the domain I-II hinge. Residues 401 and 403, which occupy the αA/B groove, are highlighted as green spheres, showing that they cement the base of the domain II trimeric interaction above the flexible hinge region. The left and right panels are close-ups of the two boxed regions viewed down the trimer axis, showing in gray the dynamically exposed areas of the trimer interface.

C Open-book view of the sE trimer (PDB 1URZ) in surface representation, with two subunits on the left and the missing "open" subunit on the right. The surfaces spanned by the HDX-$a$, $b$, $c$, and $d$ peptides are colored gray. The non-crystallographic three-fold axis is indicated by a vertical black line.

D Close-up of regions HDX-$b$ and HDX-$d$ (in gray and highlighted with ovals) involved in domain I/domain III inter-chain interactions.

Data information: Six labeling replicates (technical replicates) were done for each protein. The statistical analysis package in Mass Spec Studio v1.3.2 software was used to mine the HDX data and identify regions of perturbed labeling [62].

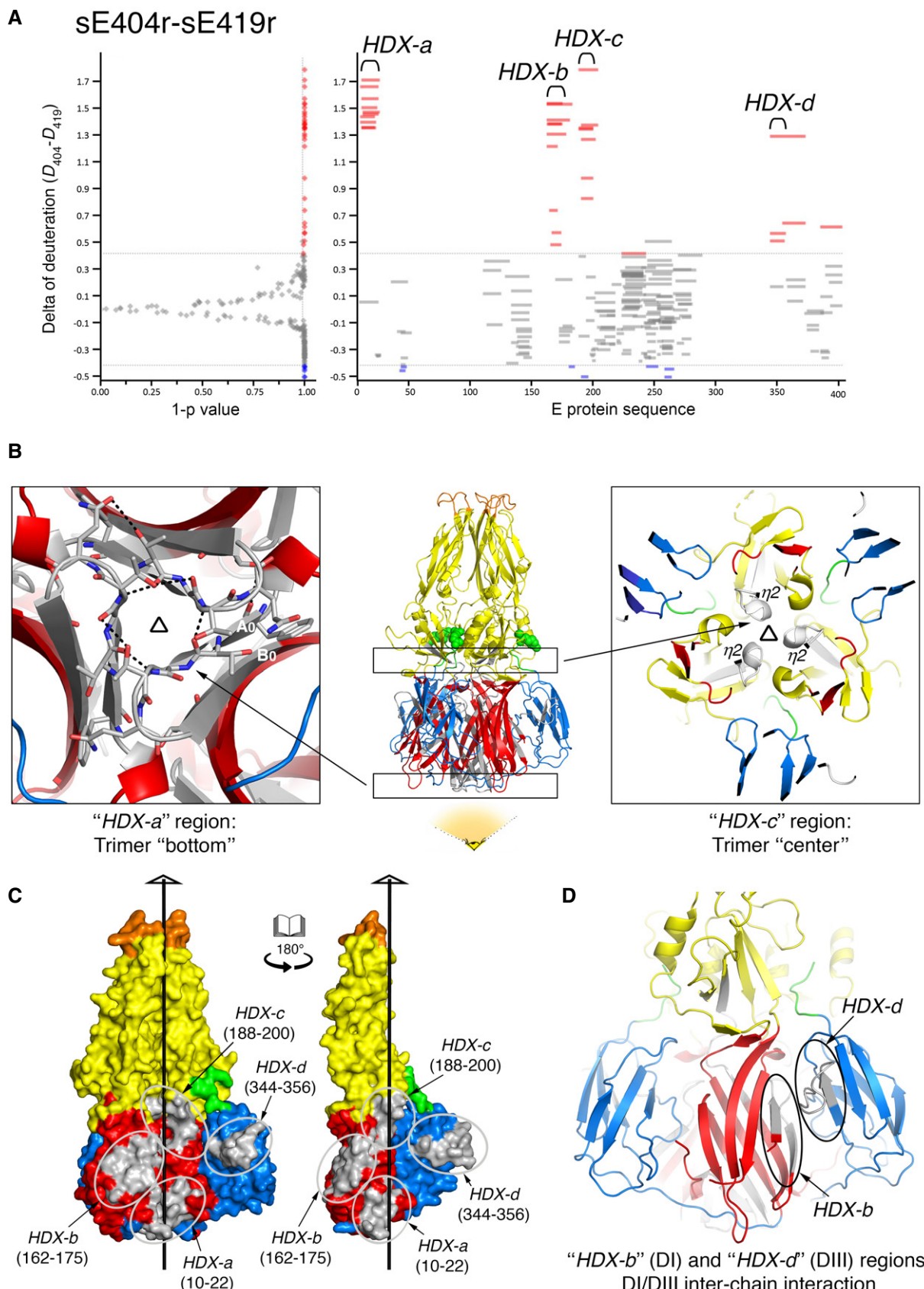

**Figure 5.**

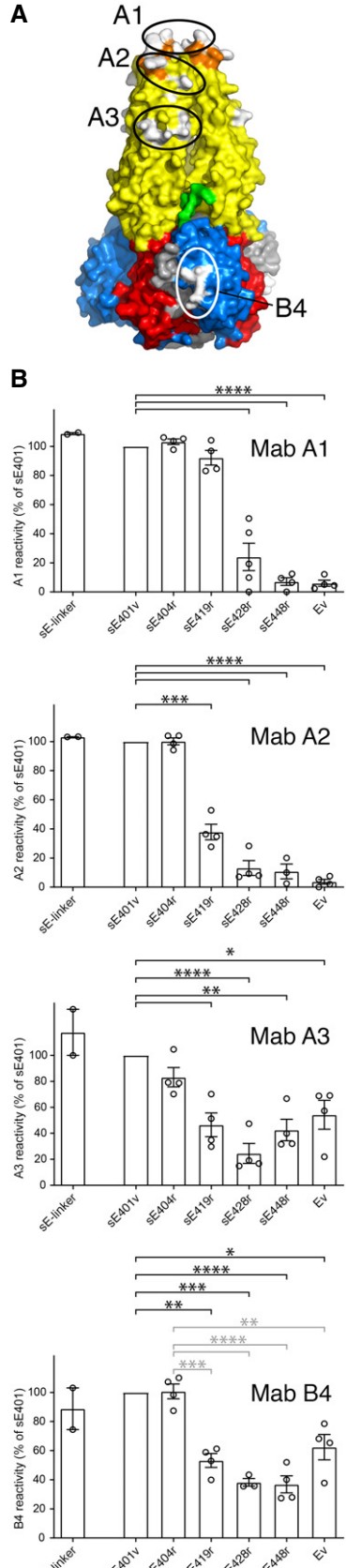

**A**  A1  A2  A3  B4

**B**  Mab A1  Mab A2  Mab A3  Mab B4

**Figure 6. Mab binding data and corresponding epitopes.**

A  Binding sites of mabs (highlighted with ovals) were mapped by mutational analysis (A1 and A2) or by neutralization escape (A3 and B4). The residues affecting binding are represented as white patches on the surface of the sE trimer (PDB 1URZ) color-coded as domains. The *a–d* surface segments identified by the HDX-MS experiments are shown as gray patches.

B  Blocking ELISA with mabs A1, A2, A3, and B4. Serial dilutions of the trimers were pre-incubated with a predetermined concentration of the respective mabs. The fraction of mab not blocked by the antigen was detected in ELISA with TBEV-coated plates (Materials and Methods). The bar charts show the results expressed relative to the reactivity with the sE401v trimer. Data are from at least three independent experiments.

Data information: Data shown represent the means ± range (sE-linker, n = 2 biological replicates) or ± standard error of the mean (n = 3–6 biological replicates, all other trimers). Statistical significance (excluding the sE-linker) was determined using one-way ANOVA with Dunnett's multiple-comparison test (n.s, not significant; ****$P < 0.0001$; ***$P < 0.001$; **$P < 0.01$; *$P < 0.05$). Black asterisks indicate significant differences relative to the sE401v trimer; gray asterisks in the bottom panel indicate significant differences to sE404r. Only comparisons described in the text are shown, and complete statistical analyses are summarized in Appendix Table S3.

protected in a static molecule (Fig 5), indicating a long-distance effect of the zippering reaction. The observed allosteric effect therefore suggests that in early stages of the fusion reaction, when the complete hairpin has not yet zippered up, the intermediate E trimer is highly dynamic and can sample multiple conformations. Consistent with this interpretation, previous studies had reported stabilization of the core E trimer by stem elements including H1, H2, and CS sequences. These studies involved reconstitution experiments using DENV2 core E trimers made of only domains I and II, which were stabilized by adding exogenous domain III + stem of different lengths [35].

Our analyses suggest a revised model of the flavivirus post-fusion E trimer that is supported by the structure of the only class II arboviral fusion protein for which the stem has been traced up to the FL in the post-fusion trimer: the Rift Valley Fever Virus (RVFV) glycoprotein Gc (Fig 7) [36]. Comparison with the RVFV Gc trimer suggests that the most C-terminal residue resolved in the flavivirus sE trimer crystal structures (i.e., residue 404 in TBEV) is approximately 15 amino acids away from the FL (Fig 7). As in Gc, this 15-residue segment can span the required distance (~ 44 Å) in an extended conformation. In line with the differences observed between the sE404r and sE419r trimers with respect to thermostability and deuterium exchange (Figs 4B and 5A), the interactions established by the non-resolved stem residues with domain II contribute to trimer stability and reduce its dynamic behavior. In the RVFV Gc structure, the stem was further shown to contact a glycerophospholipid (GPL) head group inserted in a cavity underneath the FL, an interaction that appeared to stabilize the FL-proximal portion of the stem on the sides of the trimer [36]. Such a lipid head binding pocket was proposed to be present in the flavivirus E protein as well [36]. It is possible that as in RVFV Gc, only the presence of a bound GPL head group stabilizing the FL-proximal part of the stem would allow to resolve the structure of a fully zippered E trimer.

In our model of the post-fusion E trimer, the downstream H3 segment does not interact with the trimer core and remains lying on the membrane's outer leaflet as an amphipathic helix during the

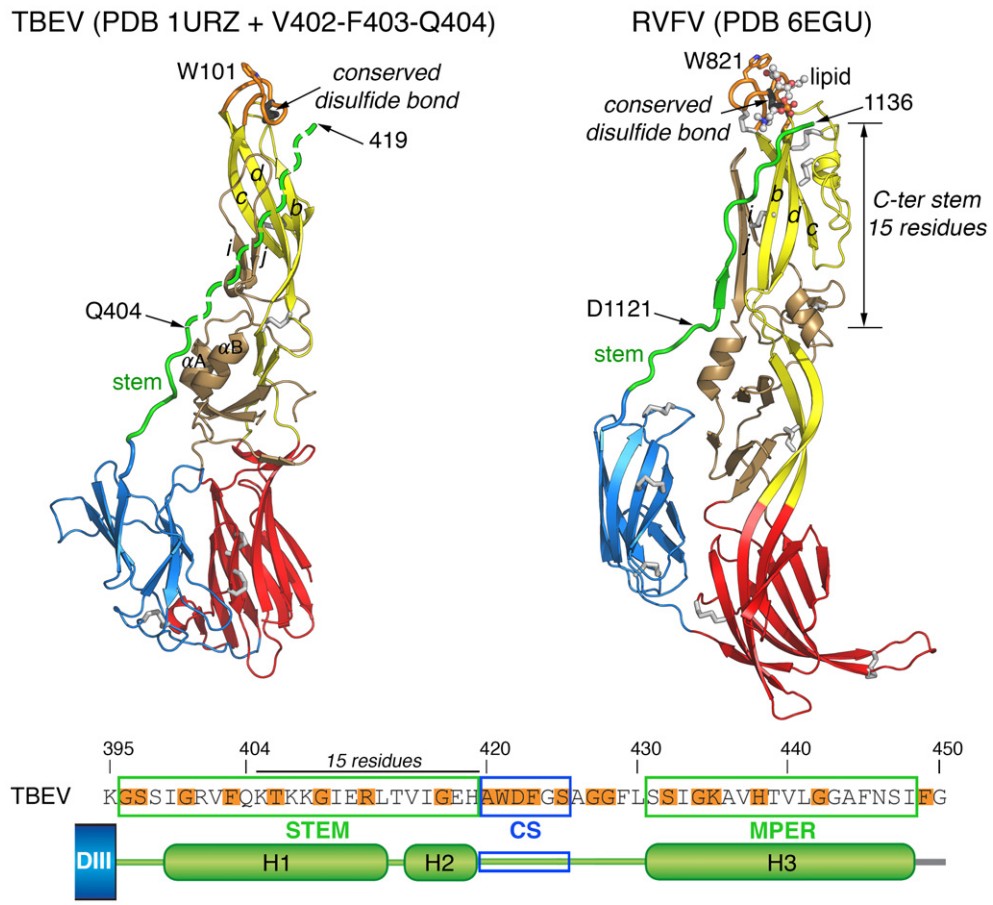

**Figure 7. Comparison of the class II post-fusion structures of TBEV sE and RVFV Gc.**

One protomer extracted from the post-fusion trimer of TBEV sE (PDB 1URZ [15] + stem residues 402–404 modeled from the sE-linker* structure) displayed next to a protomer extracted from the complete trimer of hairpins of the RVFV Gc (PDB 6EGU [36]) class II fusion protein. Both proteins are color-coded by domains as in Fig 1 but with the two segments that make up domain II distinguished in yellow and brown. An arrow at the top marks a disulfide bond conserved across all class II viral fusion proteins (shown in black). In RVFV Gc, the stem reaches the FL and its C-terminal end is stabilized by interactions with a GPL head group (shown as ball-and-stick) bound in a cavity underneath the FL, with its glycerol moiety interacting with the conserved disulfide bond. The most C-terminal residue visualized in the TBEV sE-linker* structure, Gln404, is about 15 amino acids short of reaching the FL, which would correspond roughly to position 419 at the beginning of the CS. The location of CS within the stem is indicated in the panel below.

fusion reaction (as diagrammed in Fig 8A, left panel, and 8B). In the RVFV Gc trimer, the conserved region that comes into contact with the FL at the end of the stem (which is conserved in insect-borne phleboviruses and would correspond to the CS of flavivirus E) is also followed by an amphipathic helix before reaching the TM anchor (Fig EV4, right panel). This amphipathic helix, termed the membrane-proximal external region (MPER), does not make interactions with the body of the RVFV Gc post-fusion trimer, as in the model proposed here for the H3 helix of flavivirus E. The membrane into which H3 is embedded is initially the viral membrane and becomes the fused membrane upon lipid merger (Fig 8B). H3 is potentially important in catalyzing membrane merger, because its helical conformation may allow for an "extensible" polypeptide stretch to accompany the conformational gymnastics of the stem during the fusogenic conformational change, while simultaneously destabilizing the viral lipid bilayer (Fig 8B). Similar amphipathic MPERs have been described for viral fusion proteins of different structural classes, such as the class I gp41 glycoprotein of HIV and the class III gB glycoprotein of herpes simplex virus [37,38].

Our studies with E-protein trimers carrying sequential extensions of the stem (mimicking late intermediate forms occurring during the flavivirus membrane fusion process, in which the stem-zippering reaction is still incomplete, Fig 8B, steps iv and v) provide evidence for highly dynamic fusion intermediates. It has to be noted that our experiments were performed with E trimers lacking the membrane-anchoring segment. However, during the initial and intermediate stages of fusion the ectodomains are loosely tethered to the viral membrane and would thus behave rather independently of the TM segments—similar to the truncated E trimers in our analyses. In contrast, with respect to the target membrane, the three tips of domain II may have less freedom to move, and hence, the required hinging motion about the domain I/II interface becomes important. The significantly higher deuterium exchange rates in surfaces that are buried in the structures of the post-fusion trimer suggest rapid dissociation and re-association in these intermediate forms. The strong anchoring of residues 401/403 of the stem inserted in the αA/B groove introduces additional inter-subunit contacts (Fig 2B) that keep the domain II tips together in

the trimer, preventing its full dissociation as the domain I/III moiety "breathes" about the hinge between domains I and II (Fig 8B, steps iv and v). In contrast, a substantial component of E breathing on mature particles [20,39,40], in which the domain I/III moiety is constrained by its interactions on virions, involves the domain II tip and exposes the FL (Fig 8B, step i). In both cases, however, breathing appears to involve flexing about the hinge between domains I and II.

Stem-derived peptides spanning the CS and H3 helix of DENV2 E had been shown to bind to stemless DENV2 E trimers and to inhibit membrane fusion as well as DENV2 infectivity [21,23]. The binding of these peptides could be mediated by the CS, which according to our model would interact with the FLs in the post-fusion E trimer (Fig 8). Since we did not observe an increase in trimer stability introduced by the presence of the CS in our constructs (Fig 4B), we conclude that these interactions must be relatively weak. This conclusion appears difficult to reconcile with the previously reported data on the binding and fusion-inhibiting properties of peptides containing CS and H3. The most likely explanation for these seemingly discrepant data could be that the peptides multimerize through interactions of their amphipathic H3 helices, hiding their hydrophobic side away from solvent, as diagrammed in Fig 8A, second panel. Indeed, the oligomeric state of the peptides used in those experiments had not been analyzed in the reported studies. Presentation of the CS in a multimeric form would enhance the avidity of the peptide for the three FLs at the tip of the E trimer, allowing sufficiently stable contacts to be formed. In hindsight, trimerization via the H3 moiety in our sE-linker* construct, in which the 8-aa linker is too short to allow H3 to be positioned beyond the tip of the trimer (as diagrammed in Fig 8A, third and fourth panels), could force the E tip to come apart and may have introduced the observed disorder in the crystal structure.

Our results suggest an important role of the αA/B groove of domain II in "anchoring" the stem at the sides of the trimer, midway between the end of domain III and the FL (Fig 8A, left panel). The polypeptide segment downstream (residues 404–419)

can then exert tension pulling the H3 helices together with the outer leaflet of the viral membrane, bringing it closer to the FLs. The accepted model for the fusogenic conformational change of membrane-bound E trimers postulates tilting of the trimers with respect to the viral membrane. The resulting asymmetry will result in the three E subunits undergoing significantly different pulling forces on their TM segments, depending on whether they are on the viral membrane proximal or distal side of the tilted trimer (Fig 8B, steps iv and v). We propose that these asymmetric pulling forces re-orient the trimer with one of the stems closest to the viral membrane. This arrangement implies one subunit toward the target membrane ("U", for upper subunit) and the other two toward the viral membrane ("L", for lower subunits) (Fig 8B, step v). The proposed asymmetry thus introduces an L/L stem (running in between the two L subunits) lying against the viral membrane and facilitating its complete zippering, while zippering of the U/L stems would lag behind (Fig 8B, step v, inset). In this arrangement, the H3 helix and TM segments of the U/L stems would further destabilize the viral membrane by pulling it toward both sides of the trimer to complete the zippering. The increased stability of the fully zippered trimer would help the directionality of the reaction, making it less favorable to proceed backwards. We note that the initial insertion of E into the target membrane via the FLs implies the presence of a lipid head group in the proposed GPL-binding pocket at the tip of domain II [36], which we postulate would strengthen the interaction with the CS. In the absence of a lipid head group, the CS/FL interaction appears not to be strong enough to stabilize the stem at the sides of the E trimer, explaining the observed disorder in the crystals of the DENV1 sE trimer [19].

The breathing behavior of flavivirus E has been extensively documented with respect to its interactions with antibodies in the pre-fusion conformation [20,39,40]. Our observations of the dynamics of incompletely zippered post-fusion trimers show that E breathing is not restricted to the transient exposure of cryptic sites on mature virions, such as the conserved FL, which in the case of dengue virus appears to be linked to antibody-dependent enhancement of infection [20]. The results presented here now

**Figure 8. New model for the flavivirus post-fusion trimer and implications for the membrane fusion process.**

A Postulated organization of the amphipathic H3 helix in the post-fusion trimer. Left panel: proposed arrangement of the H3 helix in our post-fusion trimer model, lying flat on the fused membrane (in light orange) and projecting the TM segments (not drawn) away from the trimer axis. Second panel: potential conformation of the sE448r trimer, with the H3 helices trimerizing via their hydrophobic surfaces. Third panel: the sE-linker construct. As the linker is not long enough (only eight residues), the H3 helix cannot trimerize as in the full-stem sE448r trimer. It would be exposed on the sides of the trimer, potentially leading to aggregation. Right-hand panel: the sE-linker* did not lead to aggregation, and at high concentrations, it crystallized in a form that had the tip disordered. Trimerization of the three H3 helices in the trimer may have caused this effect, as they must displace the trimer tips in order to oligomerize. This is a further measure of the highly dynamic breathing behavior of the E trimers, as the sequence required to complete the stem zipper was absent from this construct.

B (i) Schematic of the viral surface in the mature particle, with E and M in side view, color-coded as in Fig 1. The breathing behavior within and between E dimers, transiently exposing the fusion loop and regions normally buried at the dimer–dimer interface, is indicated by curved lines. A vertical arrow points to the hinge between domains I and II, a region that supports E-protein breathing. (ii). Dissociation of E dimers in the acidic milieu of an endosome, exposing the FLs and allowing their insertion into the endosomal membrane. (iii). Upon fusion loop insertion, the E proteins form "extended intermediate" trimers bridging the two membranes. Curved blue arrows indicate the subsequent relocation of domain III, in blue, to the sides of the trimer to initiate the "hairpin" conformation of the individual subunits of the post-fusion trimer. (iv). Upon relocation of domain III, the upstream part of the stem becomes anchored to the αA/B groove (red circle) through residues 399–403 (see also Figs 2B and EV1) to form an additional intermediate, a dynamic stage ("breathing trimer", indicated by curved lines as in panel (A), with black arrows pointing to the hinge region). The extensible H3 segment (which remains membrane-associated) is pulled toward domain II along with the TM segments (green broken double arrows). A blue square marks the CS (labeled). Only one stem-anchor region per trimer is shown for clarity. (v). The asymmetric TM "pulling" forces are proposed to orient the trimer with one subunit closest to the target membrane ("U", for upper, right panel), and the two lower (L) subunits toward the viral membrane. The spatial geometry suggests that the stem between the two L subunits will zipper up first to reach the FLs, while the other two remain partially zippered. Because the three-dimensional membrane deformation is difficult to represent in two dimensions, the view toward the fully zippered L/L stem (white arrow) is schematically shown in the inset (the corresponding TM segment is not represented). (vi). The post-fusion E trimer and formation of a fusion pore. The stem is fully zippered along the three sides of the trimer, with all three CSs interacting with the FL and H3 bound to the fused membrane (although only one is shown, for clarity).

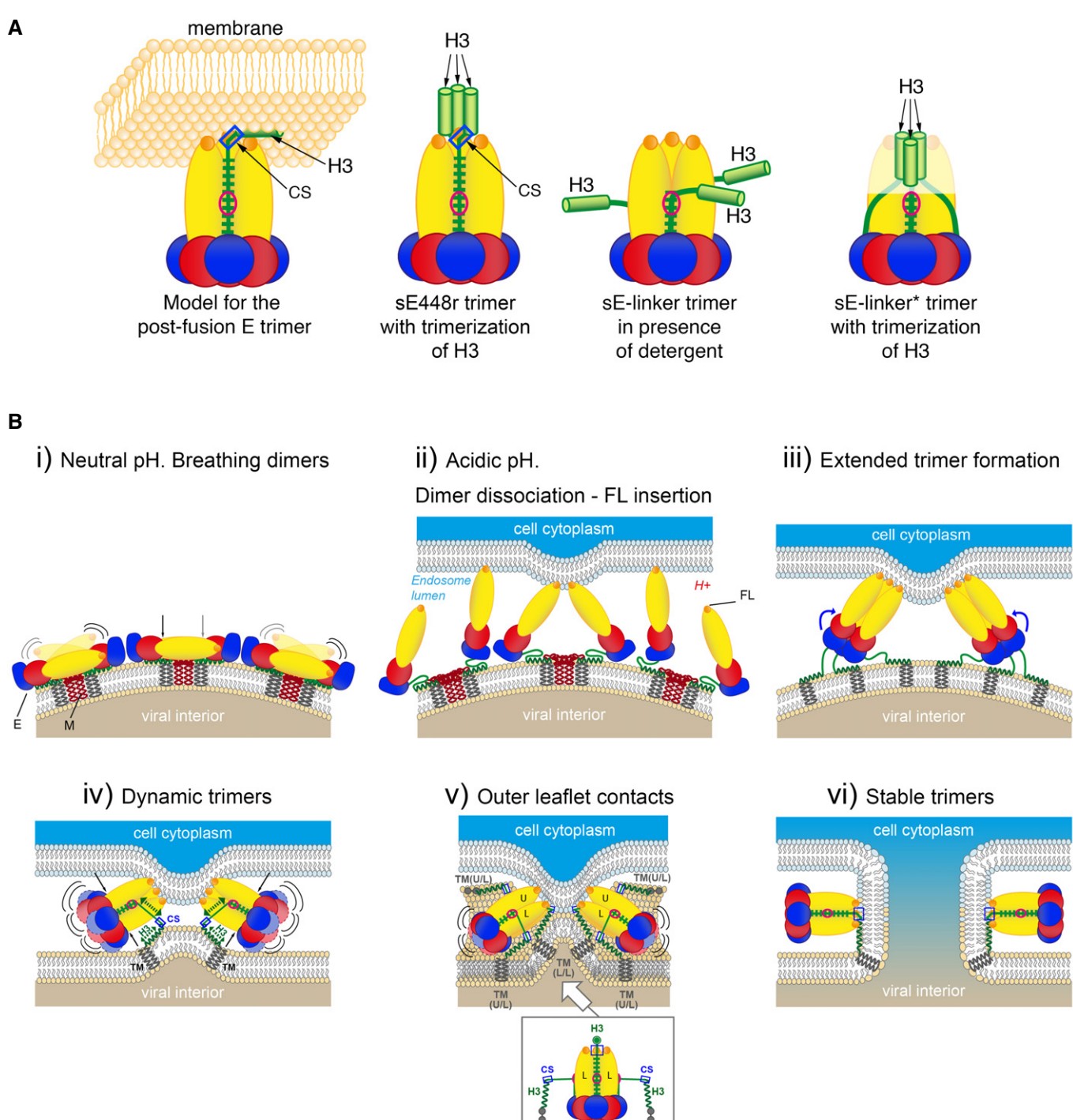

**A**

membrane

H3

CS

Model for the
post-fusion E trimer

H3

CS

sE448r trimer
with trimerization
of H3

H3

H3

H3

sE-linker trimer
in presence
of detergent

H3

sE-linker* trimer
with trimerization
of H3

**B**

**i)** Neutral pH. Breathing dimers

E

M

viral interior

**ii)** Acidic pH.

Dimer dissociation - FL insertion

cell cytoplasm

*Endosome
lumen*

*H+*

FL

viral interior

**iii)** Extended trimer formation

cell cytoplasm

viral interior

**iv)** Dynamic trimers

cell cytoplasm

CS

H3

H3

TM

TM

viral interior

**v)** Outer leaflet contacts

cell cytoplasm

TM(U/L)

TM(U/L)

U

L

TM
(L/L)

TM
(U/L)

TM
(U/L)

H3

CS

CS

H3

L

L

H3

TM
(U/L)

TM
(U/L)

**vi)** Stable trimers

cell cytoplasm

viral interior

Figure 8.

identify flavivirus envelope protein breathing as functionally important also for the molecular gymnastics required to drive membrane fusion. In both events, a substantial component of the breathing is supported by the hinge between domains I and II. These data further point to additional vulnerability sites, such as the αA/B groove of the E protein, as a target for the development of efficient anti-flavivirus molecules.

## Materials and Methods

### Cloning procedures

Recombinant sE proteins (sE400r, sE401r, sE412r, sE419r, sE428r, sE448r, sE-linker) were derived from TBEV strain Neudoerfl (GenBank accession number U27495). All antigens were expressed

in *Drosophila* Schneider 2 (S2) cells (Invitrogen) with an enterokinase cleavage site and a strep-tag. The pT389-sE400 [41], pT389-sE419 [28], and pT389-sE448 [28] expression plasmids were already available.

The pT389-sE401 expression plasmid was generated by mutagenesis of the pT389-sE400 clone (GeneArt site-directed mutagenesis system, Invitrogen) following the manufacturer's instructions.

For the generation of the plasmids pT389-sE404, sE408, sE412, and sE428, the stem elements present in pT389-sE448, flanked by suitable restriction sites, were amplified by PCR. The PCR products and the initial plasmid were cleaved with the enzymes SnaBI and ApaI (Fermentas) following the manufacturer's instructions. For the p389T-sE 404-linker-448 (pT389-sE-linker) construct, a plasmid containing an insert with the codons for the linker was ordered and used for cloning (Thermo Fisher Scientific GENEART). This plasmid and the pT389-sE448 construct were cleaved with the enzymes Bsp120I and PasI (Fermentas). DNA fragments and plasmids were ligated with a T4 ligase (Fermentas) as recommended in the manufacturer's manual.

A point mutation in the FL of E, Trp 101 to His (W101H), was introduced into the p389T-sE-linker plasmid by site-directed mutagenesis (GeneArt site-directed mutagenesis system, Invitrogen) following the manufacturer's instructions to make the sE-linker* protein used in the structural studies.

## Expression and purification of recombinant proteins

The recombinant proteins were expressed in *Drosophila* S2 cells as described previously [41–43]. Briefly, calcium phosphate transfection of cells was performed with the different expression plasmids and a plasmid containing a blasticidin resistance gene for selection according to the manufacturer's protocol (Invitrogen). After transfection, the medium was supplemented with 25 μg/ml blasticidin (Fisher Scientific) to select stably transfected cells.

For protein expression, cells were transferred into serum-free medium (Lonza) containing 10 μg/ml blasticidin, and the cells were cultivated under shaking conditions at 28°C. Expression of the recombinant proteins was induced by the addition of 1 mM CuSO$_4$. After incubation of 7–10 days, the supernatant was harvested, clarified, and concentrated by ultrafiltration using Vivaflow 200 (30 MWCO, Sartorius). The strep-tagged proteins were purified by affinity chromatography using Strep-Tactin columns (IBA GmbH) according to the manufacturer's instructions. Recombinant proteins were eluted with 100 mM Tris–HCl, 150 mM NaCl, and 1 mM EDTA containing 10 mM desthiobiotin. The pooled fractions were then concentrated and buffer-exchanged to TAN buffer pH 8.0 (50 mM triethanolamine, 100 mM NaCl) using Vivaspin® 6 centrifugal concentrators (30 MCWO, Sartorius).

Protein concentrations were determined with the Pierce BCA Protein Assay (Thermo Fisher Scientific) following the manufacturer's protocol. Purity of the proteins was verified by 10–12% sodium dodecyl sulfate–polyacrylamide gel electrophoresis (SDS–PAGE) according to Laemmli [44].

## Preparation of the virion-derived sE401v protein

sE401v was generated by limited trypsin digestion of purified virions at 4°C. Residual virus particles were removed by ultracentrifugation, and purification was performed by anion exchange chromatography as described previously [45].

## Cleavage of the strep-tag

To remove the strep-tag, 250 μg of recombinant protein was incubated with 25 units enterokinase (EKmax, Invitrogen) for 30 min at 4°C as described in [28]. To block protease activity, aprotinin was added at a ratio of 1 μg aprotinin to one Unit EKmax.

## Crystallization and X-ray structure determination

The sE-linker* protein was further purified by size-exclusion chromatography (SEC) using a Superdex 200 Increase 10/300 GL column (GE Healthcare) in 50 mM Tris–HCl pH 8, 500 mM NaCl buffer. Prior to crystallization, the purified protein was concentrated to approximately 0.42 mg/ml and buffer-exchanged to 15 mM Tris–HCl pH 8, 150 mM NaCl. Crystallization conditions were screened in 400 nl sitting drops formed by mixing equal volumes of the protein and reservoir solution in 96-well Greiner plates with a Mosquito robot and monitored by Rock-Imager according to the procedures described by Weber *et al* [46]. Initial crystals were optimized using a robotized system (Matrix Maker and Mosquito setups). Diffraction data were collected at several beamlines at synchrotrons SOLEIL (St Aubin, France) and the Swiss Light Source (Villigen, Switzerland). The data sets were indexed, integrated, scaled, and merged using XDS [47] and AIMLESS [48]. Molecular replacement was done using the TBEV sE trimer structure (PDB 1URZ, [15]) with PHASER [49]. These programs are part of the CCP4 suite [50]. The asymmetric unit of all the rhombohedral crystals contained one extended sE-linker* trimer, with the cell parameters varying along the c axis from 118.6 to 123.5 Å, together with a solvent content varying from 42.5 to 47.2%, and diffraction to resolutions reaching 2.33 to 2.6Å, respectively. The structures were determined from several of these crystals in the hope to find one with less disorder. The final data set used to analyze the structure had the least anisotropic resolution and the longer cell edges ($a$ = 169.4 Å, $b$ = 169.4 Å, $c$ = 123.5 Å) and was from a crystal grown in 19% isopropanol, 100 mM Tris-sodium citrate pH 5.6, 19% PEG 4000, and 5% glycerol. This crystal was plunged directly into liquid nitrogen and the data set at 2.57 Å resolution was collected on a PILATUS 6MF detector at the X06SA beamline (SLS synchrotron). Data collection and refinement statistics for this crystal are presented in Appendix Table S1. All the resulting atomic models were refined with BUSTER/TNT [51] alternating with manual corrections with COOT [52]. Refinement was carried out imposing 3-fold non-crystallographic symmetry constraints and parametrization describing translation, libration, and screw-rotation to model anisotropic displacements (TLS). Figures of structures were prepared using the PyMOL Molecular Graphics System, Version 2.1.0 (Schrödinger LLC).

## Generation of liposomes

Liposome production was carried out as described in [53]. 1-Cholesterol (Sigma), phosphatidylcholine (PC), and phosphatidylethanolamine (PE) (both from Avanti Polar Lipids) dissolved in chloroform at a molar ratio of 2:1:1 were dried to a thin film using a rotary

evaporator followed by applying a high vacuum for 1.5 h. The liposomes were resuspended in liposome buffer (10 mM triethanolamine, 140 mM NaCl, pH 8.0), and five cycles of freeze-and-thaw were performed. Liposomes were extruded through two polycarbonate membranes with a pore size of 200 nm using the Liposofast syringe-type extruder (Avestin, Ottawa, Ontario, Canada).

### Trimer preparations

Recombinant proteins after enterokinase cleavage (sE400r, sE401r, sE404r, sE408r, sE412r, sE419r, sE428r) as well as the virion-derived sE401v [45] were converted into trimers under acidic pH conditions in the presence of liposomes as described previously [28–30]. Fifteen nanomoles of lipids per microgram sE was mixed and acidified by the addition of 350 mM morpholineethanolsulfonic acid (MES). After a 30-min incubation at pH 5.4 and 37°C, the mixture was back-neutralized and diluted to a final sucrose concentration of 20% (wt/wt). This solution was applied to a 50% (wt/wt) sucrose cushion and overlaid with 15% (wt/wt) and 5% (wt/wt) sucrose. After ultracentrifugation (Beckman SW 55; 240,000 $g$; 90 min; 4°C), the trimer-containing top fractions were solubilized with 1.5% $n$-octylglucoside (nOG) for 1 h followed by ultrafiltration (Vivaspin 20; 100,000 MWCO PES; Sartorius) to remove lipids. The trimer preparations were then applied to Strep-Tactin spin columns (IBA GmbH) to remove residual tagged trimers, which was controlled by Western blotting [54] using a Strep-Tactin-conjugated horseradish peroxidase (IBA GmbH, Göttingen, Germany. #2-1502-001).

sE448r proteins, which were already trimers in the cell culture supernatant, were purified by rate zonal centrifugation after solubilization with 1% TX-100 as described previously [28].

Full-length E trimers (Ev) were produced by solubilization of purified low-pH-treated virus with 1% Triton X-100 followed by rate zonal centrifugation [28,30].

### Quantitative four-layer ELISA

E protein was quantified by a 4-layer enzyme-linked immunosorbent assay (ELISA) as described previously [55]. Briefly, microtiter plates (NuncU Maxisorp) were coated with a TBEV-specific polyclonal guinea pig serum (obtained from the Core Unit of Biomedical Research, Division of Laboratory Animal Science and Genetics, Medical University of Vienna, Himberg, Austria) in carbonate buffer pH 9.6 for 48 h at 4°C. Protein samples and a TBEV protein standard (purified virus or E protein) were incubated in the presence of 0.4% sodium dodecyl sulfate (SDS) for 30 min at 65°C. Serial dilutions of the samples and the standard were added and incubated for 90 min at 37°C. A polyclonal rabbit anti-TBEV serum (obtained from the Core Unit of Biomedical Research, Division of Laboratory Animal Science and Genetics, Medical University of Vienna, Himberg, Austria) was then added for 1 h at 37°C followed by a 1-h incubation with a peroxidase-conjugated donkey anti-rabbit immunoglobulin G (DAR-POX, Amersham. #NA934V). Data were evaluated with the software Gen5 Data Analysis using the TBEV protein standard.

Both the anti-TBEV guinea pig and rabbit serum were generated by immunization with purified formalin-inactivated TBEV.

### Sedimentation analyses and dissociation of E and M proteins

Seventy microgram of purified TBEV [56] was incubated for 10 min at 37°C at pH 6.0 or pH 8.0. The samples were back-neutralized, solubilized with 0.5% Triton X-100, and subjected to rate zonal gradient centrifugation (7–20% sucrose in TAN buffer pH 8.0) in the presence of 0.1% Triton X-100 [57]. After centrifugation for 20 h at 180,000 $g$ and 15°C (SW40 rotor, Beckman), the gradients were fractionated. The proteins in the gradient fractions were precipitated with trichloroacetic acid (TCA) and analyzed by SDS–PAGE according to Laemmli [44]. The gels were stained with Coomassie Blue and evaluated densitometrically.

### Sedimentation analyses and thermostability

Conversion of dimers into trimers was confirmed by sedimentation analysis as described in [28,30,53]. Solubilized trimer preparations were applied to 7–20% sucrose gradients in TAN buffer pH8.0 containing 0.1% Triton X-100. After centrifugation for 20 h at 180,000 $g$ and 15°C (SW40 rotor, Beckman), the gradients were fractionated, and the amount of E protein per fraction was determined by a quantitative four-layer ELISA as described above.

To analyze the thermostability of the different trimers, the proteins were incubated for 10 min at 37 or 70°C, cooled down on ice, and then subjected to sedimentation in sucrose gradients as described above.

### Blocking ELISA

Reactivity of mabs A1, A2, A3, and B4 [31–34,58] with the different trimer preparations was analyzed in blocking ELISAs as described previously [30]. 1 μg trimer was pre-incubated with a fixed dilution of mab for 90 min at 37°C. The mixture was then transferred to microtiter plates coated with formalin-inactivated TBEV (1 μg/ml) and incubated for 60 min at 37°C. Detection of antibodies not blocked by the antigen in solution was performed by a peroxidase-conjugated rabbit anti-mouse immunoglobulin G (RAM-POX, Nordic MUbio, #RAM/IgG (H+L)/PO) and enzymatic reaction as described above for the quantitative four-layer ELISA. The results are expressed as percent reactivity of mab, calculated by dividing the absorbance in the presence of blocking antigen by the absorbance in the absence of blocking antigen ($A_{490\ nm}$ in the presence of mab/$A_{490\ nm}$ in the absence of mab) × 100.

### Chemical cross-linking with DMS

Proteins were chemically cross-linked as described previously [57,59]. 3 μg of protein was incubated with 10 mM dimethylsuberimidate (DMS, Pierce) for 30 min at room temperature, and the reaction was stopped with 10 mM ethanolamine. The cross-linked samples were precipitated with TCA and subjected to SDS–PAGE using 5% polyacrylamide gels under non-reducing conditions as described in [60]. Staining was performed with Bio-Safe Coomassie G-250 Stain (Bio-Rad Laboratories).

### Peptide mapping and identification using nanoLC-MS/MS

Twenty picomoles of each protein was digested for 2 min in 0.75% formic acid (FA) with 2 μl of acid-activated and concentrated

secretion of the pitcher plant *Nepenthes* and immediately lyophilized to inactivate digestive enzymes. Before mass analysis, samples were resuspended in 0.1% FA. Digests were analyzed by nanoLC-MS/MS using an Ultimate 3000 Nano-HPLC system (Dionex, Thermo Scientific) coupled to the nanoelectrospray ion source of an Orbitrap Fusion Lumos mass spectrometer (Thermo Scientific). Peptides were loaded on an Acclaim PepMap C18 precolumn (C18, 3 μm, 100 Å, 5 mm, Thermo Scientific) at a flow rate of 10 μl/min of solvent A (0.1% FA (v/v) in water) for 5 min, and eluted onto an in-house packed 25 cm nano-HPLC column with C18 resin (Aeris PEPTIDE XB-C18, 1.7 μm, 100 Å, 75 μm inner diameter). Peptides were separated at 250 nl/min using a gradient of 7% to 50% of solvent B (80% acetonitrile (v/v), 0.1% FA (v/v) in water) for 30 min followed by an increase to 95% solvent B in 1 min. The column and precolumn were then washed for 10 min at 95% solvent B and reconditioned at 4% solvent B for 20 min. NanoLC-MS/MS experiments were conducted in data-dependent acquisition mode. After a MS survey scan at a resolution of 60,000 (at $m/z$ 400 in the Orbitrap), the most intense ions, above an intensity threshold of, were selected for HCD fragmentation at NCE 30 using a resolution of 30,000. Only charge states between 1 and 10 were selected, and a dynamic exclusion of 20 s was set. The FT automatic gain control (AGC) was set to $3 \times 10^6$ for MS and $3 \times 10^5$ for MS/MS experiments. NanoLC-MS/MS data were processed automatically using Mass Spec Studio v1.3.2 to identify peptides with the following parameters: peptide sizes between 4 and 40 amino acids, charge states between 1 and 5, mass accuracy of 7 ppm for both MS and MS/MS, and a false discovery rate (FDR) of 5%.

### Hydrogen–deuterium exchange mass spectrometry

Deuterium exchange was initiated by diluting aliquots of 20 pmol of protein twice with $D_2O$ during 100 s at 20°C. Samples were quenched upon mixing with an acidic solution (0.75% FA) to decrease the pH to 2.6 and immediately digested upon addition of 2 μl of acid-activated and concentrated secretion of the pitcher plant *Nepenthes*, which contains a natural cocktail of proteases [61]. Digested samples were immediately injected into a cooled nanoACQUITY UPLC HDX system (Waters) maintained at 4°C. The generated peptides were trapped and desalted for 2 min onto a C18 trap column (Kinetex® EVO C18, 2.6 μm, 100 Å, 2.1 × 20 mm, Phenomenex) at a flow rate of 100 μl/min of solvent A (0.15% FA (v/v) in water), and then separated at 70 μl/min by a linear gradient from 10 to 25% of solvent B (0.15% FA (v/v) in acetonitrile) in 8 min followed by an increase from 25 to 60% of solvent B in 2 min using a Kinetex® EVO C18 analytical column (1.7 μm, 100 Å, 1 × 100 mm, Phenomenex). The column was washed by two fast successive gradients from 5 to 95% of solvent B and re-equilibrated at 3% of solvent B for 5 min. Blank injections were performed between each run to confirm the absence of carryover. The LC flow was directed to a Synapt™ G2-Si HDMS™ mass spectrometer (Waters) equipped with a standard electrospray ionization (ESI) source. Mass spectra were acquired in positive-ion and resolution mode over the $m/z$ range of 300–2,000 with 0.5 s scan time. After 10.5 min, the flow was directed to waste in order to prevent detergent contained in samples from entering the mass spectrometer. Deuterium uptake

values were calculated for each peptide using Mass Spec Studio v1.3.2 software [62], and no adjustment was made for back-exchange.

### Statistical analyses

Statistical analyses for Figs 4B and 6B were performed with GraphPad Prism Version 8. After square-root transformation of the raw percentage data (not normalized to Ev or sE401v) and testing for normal distribution with the Shapiro–Wilk test, the data obtained with the different trimer preparations (biological replicates) were subjected to one-way ANOVA followed by Tukey's or Dunnett's multiple-comparison *post hoc* tests as indicated in the figure legends. $P$ values $< 0.05$ were considered statistically significant.

Statistical analyses for Fig 5 were performed with Mass Spec Studio v1.3.2 software [62].

### Accession numbers

TBE virus (TBEV) (GenBank accession number U27495); Dengue virus type 1 (DENV1) (GenBank accession number AB189120), type 2 (DENV2) (GenBank accession number M19197), type 3 (DENV3) (GenBank accession number AF349753), and type 4 (DENV4) (GenBank accession number AY618991); Zika virus (GenBank accession number KJ776791); Yellow fever virus (YFV) (GenBank accession number X03700); Japanese encephalitis virus (JEV) (GenBank accession number AF315119); and West Nile virus (WNV) (GenBank accession number AF206518).

## Data availability

The data produced in this study are available in the following database: The atomic coordinates and structure factors amplitudes of the sE-linker* trimer have been deposited in the RCSB Protein Data Bank (PDB, www.rcsb.org), with the accession code 6S8C (https://www.rcsb.org/structure/6S8C).

**Expanded View** for this article is available online.

### Acknowledgements

We thank Walter Holzer, Andrea Reiter, Hannes Prechler, Cornelia Hell, Mareike Grabner, and Jutta Hutecek for excellent technical assistance. We thank Ahmed Haouz and the staff from the Institut Pasteur protein crystallogenesis facility for their help with crystallization trials and the staff of the PX1 beamlines at the Swiss Light Source (Villigen, Switzerland) and at synchrotron SOLEIL (St Aubin, France) for beamline support. We acknowledge support from the Austrian Science Fund FWF (I1378-B13, P27501-B21), French ANR (grant ANR-13-ISV8-0002-01), Institut Pasteur, and CNRS.

### Author contributions

IM, M-CV, AR, MR, and JC-R performed the experiments. MCV determined the structure. IM, M-CV, MR, JC-R, FAR, FXH, and KS analyzed the data. FAR, FXH, and KS designed/conceived the project. IM, M-CV, FAR, FXH, and KS wrote the manuscript. All authors discussed the results and contributed to the final manuscript.

## Conflict of interest

The authors declare that they have no conflict of interest.

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
