## [Review Process File · EMBO Reports]

Extensive flavivirus E trimer breathing accompanies stem zippering of the post-fusion hairpin

Iris Medits, Marie-Christine Vaney, Alexander Rouvinski, Martial Rey, Julia Chamot-Rooke, Felix Rey, Franz Heinz, and Karin Stiasny

DOI: [10.15252/embr.202050069](https://doi.org/10.15252/embr.202050069)

Corresponding author(s): Karin Stiasny (karin.stiasny@meduniwien.ac.at), Franz Heinz (franz.x.heinz@meduniwien.ac.at), Felix Rey (felix.rey@pasteur.fr)

Review Timeline:

Submission Date:	20th Jan 20
Editorial Decision:	12th Feb 20
Revision Received:	10th Apr 20
Editorial Decision:	6th May 20
Revision Received:	11th May 20
Accepted:	13th May 20

Editor: Achim Breiling

Transaction Report:

Dear Dr. Stiasny,

Thank you for the submission of your research manuscript to EMBO reports. We have now received reports from the three referees that were asked to evaluate your study, which can be found at the end of this email.

As you will see, all referees think that the findings are of interest, but they also have several comments, concerns and suggestions, indicating that a revision of the manuscript is necessary to allow publication in EMBO reports. As the reports are below, and all points need to be addressed, I will not detail them here.

Given the constructive referee comments, we would like to invite you to revise your manuscript with the understanding that all referee concerns must be addressed in the revised manuscript and/or in a detailed point-by-point response. Acceptance of your manuscript will depend on a positive outcome of a second round of review. It is EMBO reports policy to allow a single round of revision only and acceptance of the manuscript will therefore depend on the completeness of your responses included in the next, final version of the manuscript.

Revised manuscripts should be submitted within three months of a request for revision; they will otherwise be treated as new submissions. Please contact me if a 3-months time frame is not sufficient so that we can discuss the revisions further.

1) a .docx formatted version of the final manuscript text (including legends for main figures, EV figures and tables), but without the figures included. Please make sure that the changes are highlighted to be clearly visible. Figure legends should be compiled at the end of the manuscript text.

2) individual production quality figure files as .eps, .tif, .jpg (one file per figure), of main figures and EV figures. Please upload these as separate, individual files upon re-submission.

The Expanded View format, which will be displayed in the main HTML of the paper in a collapsible format, has replaced the Supplementary information. You can submit up to 5 images as Expanded View. Please follow the nomenclature Figure EV1, Figure EV2 etc. The figure legend for these should be included in the main manuscript document file in a section called Expanded View Figure Legends after the main Figure Legends section. Additional Supplementary material should be supplied as a single pdf labeled Appendix. The Appendix should have page numbers and needs to include a table of content on the first page (with page numbers) and legends for all content. Please

follow the nomenclature Appendix Figure Sx, Appendix Table Sx etc. throughout the text, and also label the figures and tables according to this nomenclature.

For more details please refer to our guide to authors:

See also our guide for figure preparation:

http://wol-prod-cdn.literatunonline.com/pb-assets/embosite/EMBOPress_Figure_Guidelines_061115-1561436025777.pdf

4) a complete author checklist, which you can download from our author guidelines (<https://www.embopress.org/page/journal/14693178/authorguide>). Please insert page numbers in the checklist to indicate where the requested information can be found in the manuscript. The completed author checklist will also be part of the RPF.

Please also follow our guidelines for the use of living organisms, and the respective reporting guidelines: <http://www.embopress.org/page/journal/14693178/authorguide#livingorganisms>

5) that primary datasets produced in this study (e.g. RNA-seq, ChIP-seq, mass spec., modeling and array data) are deposited in an appropriate public database. See: <http://embor.embopress.org/authorguide#datadeposition>

The accession numbers and database should be listed in a formal "Data Availability " section (placed after Materials & Methods) that follows the model below. Please note that the Data Availability Section is restricted to new primary data that are part of this study.

Data availability

- RNA-Seq data: Gene Expression Omnibus GSE46843

(<https://www.ncbi.nlm.nih.gov/geo/query/acc.cgi?acc=GSE46843>)

- [data type]: [name of the resource] [accession number/identifier/doi] ([URL or identifiers.org/DATABASE:ACCESSION])

6) We strongly encourage the publication of original source data with the aim of making primary data more accessible and transparent to the reader. The source data will be published in a separate source data file online along with the accepted manuscript and will be linked to the relevant figure. If you would like to use this opportunity, please submit the source data (for example

scans of entire gels or blots, data points of graphs in an excel sheet, additional images, etc.) of your key experiments together with the revised manuscript. If you want to provide source data, please include size markers for scans of entire gels, label the scans with figure and panel number, and send one PDF file per figure.

8) Regarding data quantification and statistics, can you please specify, where applicable, the number "n" for how many independent experiments (biological replicates) were performed, the bars and error bars (e.g. SEM, SD) and the test used to calculate p-values in the respective figure legends. Please provide statistical testing where applicable, and also add a paragraph detailing this to the methods section. See: <http://www.embopress.org/page/journal/14693178/authorguide#statisticalanalysis>

I look forward to seeing a revised version of your manuscript when it is ready. Please let me know if you have questions or comments regarding the revision.

Yours sincerely,

Achim Breiling
Editor
EMBO Reports

Referee #1:

My enthusiasm for this paper is significant.

First, this work presents a more complete model for viral membrane fusion by a class of viruses with considerable global health significance (with implications for alphavirus and some bunyavirus entry processes as well). Fundamental knowledge of this important step in the viral replication cycle is important. This information will guide further development of countermeasures, which is an active area of study by multiple laboratories in the field (as discussed carefully in the manuscript). Second, multiple approaches are employed to detail the structural and biological importance of placement of different segments of the stem against the exterior of the trimer- this is more than just a structural study. Finally, the authors used rigorous techniques to demonstrate flavivirus trimers sample an ensemble of states as the fusion process unfolds. This has not been suggested previously and adds to a growing list of examples where the dynamics of virions/viral proteins may be critical for their biology. This element will be of very broad interest to the field.

Overall, the paper is very well-written; it was a pleasure to review. Well done.

Minor comments (suggested modifications for the text):

The authors should comment on the potential impact of the anchoring of E proteins into the virion membrane on dynamic properties of the trimers. If constrained in the membrane, how might the ensemble of states measured using the DX technique differ? While beyond the scope of this study to experimentally evaluate this, some discussion might be instructive.

What temperature was used to perform the deuterium exchange? I didn't note this in the methods section (and perhaps it may be standard). Since there is interesting biology regarding these viruses and the temperature of the hosts in which they replicate, this experimental detail would be helpful.

I found the diagrams of Figure S7A to be very helpful when visualizing the narrative of the discussion (~lines 400 or so). Given the whitespace present in Figure 8, I wonder if there is a mechanism to include them, so they make it into the PDF of the actual manuscript?

Line 451: Are all dynamics changes in virion attributable to the DI-DII hinge? This strikes me as non-conservative.

Comparisons with the RVFV trimer structure was a nice element of the manuscript.

Referee #2:

In this manuscript Medits et al. examine the flavivirus E protein as it undergoes the process of trimerization required for low-pH mediated fusion. The authors, who are leaders in the flavivirus field in structure and virus fusion, have carried out an extensive set of structural and biophysical experiments to probe the role of the "stem" region of the tick-borne encephalitis virus (TBEV) E protein. Using X-ray crystallography, chemical cross-linking, and HDX they examine the properties of various stem region constructs and the roles of the three helices and the conserved sequence (CS) in promoting fusion. They suggest based on their data that there are regions in the E protein that are required to be dynamic for the fusion process. The results presented here allow the authors to propose a modified model for this type II fusion process. The work is well done, the text and figures reflect a high attention to detail, and the results are an important contribution to our understanding of this fundamental property of flavivirus virions.

Specific comments:

Line 195 - the initial packing of the M protein into the alpha/B groove is very interesting. It later is displaced by the stem residues 401/403. M binding and later release at low pH is shown in Figure 3B. How were the dimer/trimer positions on the gradients determined? Given this switch in occupancy of this groove, can the authors discuss in more detail the potential role of the M protein for virion function?

Line 349 - The statement sounds misleading or confusing: "the C-terminal end of the flavivirus E trimer (ie residue 404 in TBEV)". This is not the C-terminus of the protein as there is the membrane anchor helices. Please clarify.

Line 395 - The trimer stability assayed that are discussed here are done in the absence of membranes. Can the authors comment more directly on the influence of membranes (or lack thereof) in their analyses?

Line 1018 - The authors describe breathing or dynamics between E domains I and II. There is additional breathing between E proteins that also occurs but as written it sounds like breathing is only of the former type.

Line 1025 - Instead of saying "the first part of the stem" please identify the helices.

Line 1037 - A movie would be better at conveying the asymmetry of this process and hopefully the authors are doing this or will consider this.

Referee #3:

Stiasny and colleagues present structural and functional data on the flavivirus E post fusion trimer with a focus on the conformation of the stem region.

Major findings are:

The crystal structure shows that the stem region of TBEV containing H1 is in an extended conformation similar to the previously published DENV sE structure. Destabilization of the M-E interaction by pH 6 induces the post fusion conformation. Truncation mutants of the stem influence trimer stability; the stem region up to residue 419 produces the same thermostability as the trimer with the complete stem.

Hydrogen-deuterium exchange mass spectroscopy showed flexibility of the stem, depending on the length. Stem truncations influence the binding mAbs.

Based on the data the authors propose a modified version for membrane fusion, where H3 stays associated with the membrane during fusion. The structural work is of high technical quality. The question what happens to the membrane proximal regions of viral fusion proteins during the fusion process is one of the last open questions in membrane fusion. The current work shows that at least the stem of TBEV contributes to trimer stability, but shows at the same time some conformational flexibility. The work thus provides some novel incremental insight in the fusion process.

Major points that need to be addressed:

The authors measured the stability of the truncated TBEV constructs by heating to 70°C and then quantifying the amount of trimers. This seems to be an odd method. If they want to describe real thermostability, it should be either determined by CD spectroscopy or the thermofluor assay.

All experiments have been performed in the absence of the transmembrane region, which itself could play an important stabilizing role. Especially the conformational flexibility of the stem is likely influenced by the TM. This should be considered in discussing the data.

Does an H3 peptide (solubilized as monomer) interact with truncated sE trimers (sE400 to sE419R)? H3 fused to MBP or GST or could be used as a bait to determine interaction with the truncated trimers. This would further confirm the model of part of the stem looping out or not.

The discussion reads very long and contains too much speculation, which is not really fully supported by the data.

Referee #1:

My enthusiasm for this paper is significant.

First, this work presents a more complete model for viral membrane fusion by a class of viruses with considerable global health significance (with implications for alphavirus and some bunyavirus entry processes as well). Fundamental knowledge of this important step in the viral replication cycle is important. This information will guide further development of countermeasures, which is an active area of study by multiple laboratories in the field (as discussed carefully in the manuscript). Second, multiple approaches are employed to detail the structural and biological importance of placement of different segments of the stem against the exterior of the trimer- this is more than just a structural study. Finally, the authors used rigorous techniques to demonstrate flavivirus trimers sample an ensemble of states as the fusion process unfolds. This has not been suggested previously and adds to a growing list of examples where the dynamics of virions/viral proteins may be critical for their biology. This element will be of very broad interest to the field.

Overall, the paper is very well-written; it was a pleasure to review. Well done.

We thank the referee for his/her positive comments!

Minor comments (suggested modifications for the text):

The authors should comment on the potential impact of the anchoring of E proteins into the virion membrane on dynamic properties of the trimers. If constrained in the membrane, how might the ensemble of states measured using the DX technique differ? While beyond the scope of this study to experimentally evaluate this, some discussion might be instructive.

As suggested by this and the two other referees, we now address this aspect in the Discussion (lines 391-397, marked-up manuscript).

It has to be noted that our experiments were performed with E trimers lacking the membrane-anchoring segment. However, during the initial and intermediate stages of fusion the ectodomains are loosely tethered to the viral membrane and would thus behave rather independently of the TM segments – similar to the truncated E trimers in our analyses. In contrast, with respect to the target membrane, the three tips of domain II may have less

freedom to move, and hence the required hinging motion about the domain I/II interface becomes important.

What temperature was used to perform the deuterium exchange? I didn't note this in the methods section (and perhaps it may be standard). Since there is interesting biology regarding these viruses and the temperature of the hosts in which they replicate, this experimental detail would be helpful.

The temperature of the labeling experiment was 20°C. This information is now included in Materials and Methods (line 665, marked-up manuscript).

I found the diagrams of Figure S7A to be very helpful when visualizing the narrative of the discussion (~lines 400 or so). Given the whitespace present in Figure 8, I wonder if there is a mechanism to include them, so they make it into the PDF of the actual manuscript?

We thank the referee for this suggestion. We have now reorganized Figure 8 to include the previous Appendix Fig S7A.

Line 451: Are all dynamics changes in virion attributable to the DI-DII hinge? This strikes me as non-conservative.

To take this comment into account, we modified the sentences addressing the DI-DII hinge in the Discussion (lines 403-406 and 467-468, marked-up manuscript) as well as the corresponding paragraph in the legend of Fig. 8B (also requested by referee 2, lines 1067-1070, marked-up manuscript), to say that “a substantial component of E breathing on mature particles” is about the DI-DII hinge.

Comparisons with the RVFV trimer structure was a nice element of the manuscript.

We thank the referee for this comment – the comparison was indeed important for establishing the bigger picture of our data.

Referee #2:

In this manuscript, Medits et al. examine the flavivirus E protein as it undergoes the process of trimerization required for low-pH mediated fusion. The authors, who are leaders in the flavivirus field in structure and virus fusion, have carried out an extensive set of structural and biophysical experiments to probe the role of the "stem" region of the tick-borne encephalitis virus (TBEV) E protein. Using X-ray crystallography, chemical cross-linking, and HDX they examine the properties of various stem region constructs and the roles of the three helices and the conserved sequence (CS) in promoting fusion. They suggest, based on their data that there are regions in the E protein that are required to be dynamic for the fusion process. The results presented here allow the authors to propose a modified model for this type II fusion process. The work is well done, the text and figures reflect a high attention to detail, and the results are an important contribution to our understanding of this fundamental property of flavivirus virions.

We thank the referee for acknowledging the importance and quality of our work.

Specific comments:

Line 195 - the initial packing of the M protein into the alpha/B groove is very interesting. It later is displaced by the stem residues 401/403. M binding and later release at low pH is shown in Figure 3B. How were the dimer/trimer positions on the gradients determined?

The identification of the dimer/trimer positions was confirmed by chemical crosslinking and these data are now included in the Figure.

Given this switch in occupancy of this groove, can the authors discuss in more detail the potential role of the M protein for virion function?

We thank the reviewing for this point. As the discussion is already long, we did not address this point there. Instead, we added a small paragraph in the Results, stating "The observed low-pH-induced dissociation of the M-E interactions in the pre-fusion form highlights a structural role of M, stabilizing the pre-fusion form of E at neutral pH. The release of the interactions of the E ectodomain with M, as well as with the E stem, thus appears essential for the E swiveling motion that projects the FL against the endosomal membrane to initiate fusion" (lines 215-219, marked-up manuscript).

Line 349 - The statement sounds misleading or confusing: "the C-terminal end of the flavivirus E trimer (ie residue 404 in TBEV)". This is not the C-terminus of the protein as there is the membrane anchor helices. Please clarify.

We thank the reviewer for spotting this ambiguous sentence. We have now amended it to read: "Comparison with the RVFV Gc trimer suggests that the most C-terminal residue resolved in the flavivirus sE trimer crystal structures (i.e., residue 404 in TBEV) is approximately 15 amino acids away from the fusion loop (Fig 7). As in Gc, this 15-residue segment can span the required distance (~ 44 Å) in an extended conformation" (lines 358-361, marked-up manuscript).

Line 395 - The trimer stability assayed that are discussed here are done in the absence of membranes. Can the authors comment more directly on the influence of membranes (or lack thereof) in their analyses?

Following this suggestion, as well as the comments by the two other referees, we now address this aspect in the Discussion (lines 391-397, marked-up manuscript).

Line 1018 - The authors describe breathing or dynamics between E domains I and II. There is additional breathing between E proteins that also occurs but as written it sounds like breathing is only of the former type.

Thank you for pointing this out. We have now amended the Figure legend to indicate that breathing can occur within and between E dimers (line 1067, marked-up manuscript).

Line 1025 - Instead of saying "the first part of the stem" please identify the helices.

We modified the sentence to specify the stem residues interacting with the trimer core (lines 1077-1078, marked-up manuscript). We have avoided referring to helices H1 and H2 here because our model indicates that this polypeptide segment is alpha-helical only in the pre-fusion conformation, and it adopts an extended conformation instead in the post-fusion form.

Line 1037 - A movie would be better at conveying the asymmetry of this process and hopefully the authors are doing this or will consider this.

We thank the reviewer for this suggestion. We have attempted to make a movie but we do not have the technical possibilities to do this. It would take a professional. Perhaps the Journal could help?

Referee #3:

Stiasny and colleagues present structural and functional data on the flavivirus E post fusion trimer with a focus on the conformation of the stem region.

Major findings are:

The crystal structure shows that the stem region of TBEV containing H1 is in an extended conformation similar to the previously published DENV sE structure. Destabilization of the M-E interaction by pH 6 induces the post fusion conformation. Truncation mutants of the stem influence trimer stability; the stem region up to residue 419 produces the same thermostability as the trimer with the complete stem.

Hydrogen-deuterium exchange mass spectroscopy showed flexibility of the stem, depending on the length. Stem truncations influence the binding mAbs.

Based on the data the authors propose a modified version for membrane fusion, where H3 stays associated with the membrane during fusion. The structural work is of high technical quality. The question what happens to the membrane proximal regions of viral fusion proteins during the fusion process is one of the last open questions in membrane fusion. The current work shows that at least the stem of TBEV contributes to trimer stability, but shows at the same time some conformational flexibility. The work thus provides some novel incremental insight in the fusion process.

We are pleased that the referee sees several major findings in our work, representing novel insights in flavivirus/class II membrane fusion. We agree that main steps of the fusogenic conformational change of the flavivirus E protein have been outlined earlier, and our contributions are “incremental” in this context. Yet the Devil is in the details, and as the reviewer states, understanding what happens with the membrane-proximal region of viral fusion proteins is still a major open question in the membrane fusion field. Our data

therefore provide novel insight into this poorly understood step of the membrane fusion process.

Major points that need to be addressed:

The authors measured the stability of the truncated TBEV constructs by heating to 70°C and then quantifying the amount of trimers. This seems to be an odd method. If they want to describe real thermostability, it should be either determined by CD spectroscopy or the thermofluor assay.

We have indeed considered alternative and more direct methods of measuring thermostability, including thermofluor assays and CD spectroscopy. However, these approaches did not turn out to be practical because of the requirement for the presence of detergent to prevent aggregation of our E trimers and the need for much higher amounts of protein than we could produce for these experiments. Because of these technical reasons, the only feasible means of measuring the thermostability was the method we used. In the revised version, we now explicitly address the reasons for using this method (lines 251-254, marked-up manuscript). Even if it appears odd to the reviewer, this procedure reproducibly quantified differences in the thermostability of the various constructs.

All experiments have been performed in the absence of the transmembrane region, which itself could play an important stabilizing role. Especially the conformational flexibility of the stem is likely influenced by the TM. This should be considered in discussing the data.

This point was also made by the two other referees, and we now address this aspect in the Discussion (lines 391-397, marked-up manuscript).

Does an H3 peptide (solubilized as monomer) interact with truncated sE trimers (sE400t to sE419R)? H3 fused to MBP or GST or could be used as a bait to determine interaction with the truncated trimers. This would further confirm the model of part of the stem looping out or not.

This is an important suggestion that we will definitely consider for follow-up studies. The amount of work this would represent, however, is beyond the scope of the present work.

The discussion reads very long and contains too much speculation, which is not really fully supported by the data.

The discussion is indeed long, but we considered important to provide an in depth assessment of the implications of our findings. We believe that the speculations we make will stimulate research to address them. As the other two reviewers found the paper interesting and easy to read - and did not comment on the length of the Discussion - we prefer to maintain most of it. We did shorten, however, the Discussion somewhat by removing the paragraph concerning small molecules inhibiting membrane fusion (lines 425-435, marked-up manuscript) to focus on the essential role of the viral-membrane-proximal region of the E protein, which has remained elusive so far.

Dear Dr. Stiasny

Thank you for the submission of your revised manuscript to our editorial offices. We have now received the reports from the three referees that were asked to re-evaluate your study, you will find below. As you will see, all referees support the publication of your study in EMBO reports. Referee #3 has 2 remaining points I ask you to address in a final revised version of the manuscript.

Further, I have these editorial requests I ask you to address:

- Per journal policy, we do not allow 'data not shown' (see on top of page 9 of your manuscript). All data referred to in the paper should be displayed in the main or Expanded View figures, or the Appendix. Thus, please add these data, or remove the statement, if these data are not essential. See:

<http://www.embopress.org/page/journal/14693178/authorguide#unpublisheddata>

- Please remove all the information (legends) regarding the Appendix from the main manuscript text. This should only be contained in the Appendix file.

- Please make sure that the data deposited at the RCSB Protein Data Bank is getting public, latest upon publication of the study.

- Finally, please find attached a word file of the manuscript text (provided by our publisher) with changes we ask you to include in your final manuscript text. But please provide your final manuscript file with track changes, in order that we can see the modifications done.

In addition I would need from you:

- a short, two-sentence summary of the manuscript
- two to three bullet points highlighting the key findings of your study
- a schematic summary figure (in jpeg or tiff format with the exact width of 550 pixels and a height of not more than 400 pixels) that can be used as a visual synopsis on our website.

Kind regards,

Achim

Achim Breiling
Editor
EMBO Reports

Referee #1:

The authors addressed the few suggestions I offered following their first submission appropriately, and in my view were also responsive to the comments of the other two referees. Very nice story.

High enthusiasm for publication.

Referee #2:

This is a revised version of the manuscript by Stasny and colleagues. The manuscript was very strong in the initial submission. The authors have now further improved the manuscript in this revised response to reviewers' questions. They have adequately addressed those queries to my satisfaction. The discussion is appropriate in length as they made some minor changes.

Referee #3:

The authors have responded to my previous concerns. Although I overlooked it in my first review a more detailed methods description is required:

- Buffer conditions for protein purification are lacking.
- Which protein concentration was used for crystallization?

Point-by-point response

Referee #1:

The authors addressed the few suggestions I offered following their first submission appropriately, and in my view were also responsive to the comments of the other two referees. Very nice story. High enthusiasm for publication.

Referee #2:

This is a revised version of the manuscript by Stiasny and colleagues. The manuscript was very strong in the initial submission. The authors have now further improved the manuscript in this revised response to reviewers' questions. They have adequately addressed those queries to my satisfaction. The discussion is appropriate in length as they made some minor changes.

Referee #3:

The authors have responded to my previous concerns. Although I overlooked it in my first review a more detailed methods description is required:

- Buffer conditions for protein purification are lacking.

As requested, we amended the description of protein purification on page 17 (lines 511-515, manuscript with track changes).

- Which protein concentration was used for crystallization?

As requested, we amended the description of crystallization on page 17 (lines 533-534, manuscript with track changes).

Karin Stiasny
Medical University of Vienna
Center for Virology
Kinderspitalgasse 15
Vienna
Austria

Dear Dr. Stiasny,

I am very pleased to accept your manuscript for publication in the next available issue of EMBO reports. Thank you for your contribution to our journal.

At the end of this email I include important information about how to proceed. Please ensure that you take the time to read the information and complete and return the necessary forms to allow us to publish your manuscript as quickly as possible.

As part of the EMBO publication's Transparent Editorial Process, EMBO reports publishes online a Review Process File to accompany accepted manuscripts. As you are aware, this File will be published in conjunction with your paper and will include the referee reports, your point-by-point response and all pertinent correspondence relating to the manuscript.

If you do NOT want this File to be published, please inform the editorial office within 2 days, if you have not done so already, otherwise the File will be published by default [contact: emboreports@embo.org]. If you do opt out, the Review Process File link will point to the following statement: "No Review Process File is available with this article, as the authors have chosen not to make the review process public in this case."

Should you be planning a Press Release on your article, please get in contact with emboreports@wiley.com as early as possible, in order to coordinate publication and release dates.

Thank you again for your contribution to EMBO reports and congratulations on a successful publication. Please consider us again in the future for your most exciting work.

Yours sincerely,

Achim Breiling
Editor
EMBO Reports

THINGS TO DO NOW:

You will receive proofs by e-mail approximately 2-3 weeks after all relevant files have been sent to

our Production Office; you should return your corrections within 2 days of receiving the proofs.

Please inform us if there is likely to be any difficulty in reaching you at the above address at that time. Failure to meet our deadlines may result in a delay of publication, or publication without your corrections.

All further communications concerning your paper should quote reference number EMBOR-2020-50069V3 and be addressed to emboreports@wiley.com.

Should you be planning a Press Release on your article, please get in contact with emboreports@wiley.com as early as possible, in order to coordinate publication and release dates.

Corresponding Author Name: Karin Stiasny

Manuscript Number: EMBOR-2020-50069V1